# A burning issue: Reviewing the socio-demographic and environmental justice aspects of the wildfire literature

**Alyssa S. Thomas**, **Francisco J. Escobedo***, **Matthew R. Sloggy**, **José J. Sánchez**

United States Department of Agriculture, Forest Service, Pacific Southwest Research Station, Riverside, California, United States of America

* Francisco.Escobedo@usda.gov

## Abstract

Larger and more severe wildfires are becoming more frequent and impacting different communities and human settlements. Much of the scientific literature and media on wildfires has focused on area of ecosystems burned and numbers of structures destroyed. Equally unprecedented, but often less reported, are the increasing socioeconomic impacts different people and communities face from wildfires. Such information seems to indicate an emerging need to account for wildfire effects on peri-urban or wildland urban interface (WUI) areas, newer socio-demographic groups, and disadvantaged communities. To address this, we reviewed the socio-demographic dimensions of the wildfire literature using an environmental justice (EJ) lens. Specifically using a literature review of wildfires, human communities, social vulnerability, and homeowner mitigation, we conducted bibliometric and statistical analyses of 299 publications. The majority of publications were from the United States, followed by Canada and Australia, and most dealt with homeowner mitigation of risk, defensible space, and fuel treatments in WUI areas. Most publications studied the direct effects of wildfire related damage. Secondary impacts such as smoke, rural and urban communities, and the role of poverty and language were less studied. Based on a proposed wildfire-relevant EJ definition, the first EJ publication was in 2004, but the term was first used as a keyword in 2018. Studies in WUI communities statistically decreased the likelihood that a publication was EJ relevant. There was a significant relationship between EJ designation and inclusion of race/ethnicity and poverty variables in the study. Complexity across the various definitions of EJ suggest that it should not be used as a quantitative or binary metric; but as a lens to better understand socio-ecological impacts to diverse communities. We present a wildfire-relevant definition to potentially guide policy formulation and account for social and environmental justice issues.

## 1. Introduction

Recent wildfires across the globe are increasingly larger, more severe, and incurring unprecedented costs [1]. In the United States alone, wildfires have become larger (i.e., mean size increased by 78% from 1992–2015), more frequent (by 12%), and the wildfire season has

**Data Availability Statement:** We have revised our data statement's URL to facilitate downloading by readers so that they can access the full bibliographic data and information for all 299

studies. The URL which now has the data uploaded and will now be included in this paper is: (https://github.com/msloggyfs/EJ_wildfire_data).

**Funding:** The author(s) received no specific funding for this work.

**Competing interests:** The authors have declared that no competing interests exist.

become longer (17%; [2]). Over 4 million hectares (ha) were burned in 2020, this being the third year to break this number, since 2015 [3]. Globally, Australia's 2019–2020 fire season was unprecedented in terms of the amount of any continental forest biome burned, with more than 23% (18.6 million ha) of Australian temperate forests lost to wildfire [4]. In recent years the town of Paradise, California, United States (US; population 22,000) and the village of Lytton, British Columbia, Canada (population 300) were both destroyed by fast-moving wildfires [5]. Chile also experienced its largest wildfire in the country's history in 2017, with over 0.5 million ha burned and 11 deaths [6].

In the growing body of scientific literature on wildfires, as well as in the media, both the foci and metrics have been on the surface area of ecosystems burned and the number of structures destroyed. Equally unprecedented, and often less reported, are the increasing socioeconomic impacts and diversity of people and communities being affected [7,8]. For example, the 2019–2020 Australian wildfires were the most expensive to-date for that country with total costs of US$4.5 billion [9]. In California, the estimated economic impacts from lost infrastructure and homes, health costs, and other losses from overall economic and business disruptions was US$148.5 billion in 2018, or roughly 1.5% of the state's GDP [10]. Elsewhere, in Greece, 800 families had their forest-based livelihoods (e.g., resin, honey, olive oil) destroyed in the wildfires on the island of Evia in 2021 [11]. Another recent concern is increased occurrences of human fatalities, particularly in recent peri-urban wildfires in California and Australia that claimed the lives of dozens of civilian non-firefighters [12,13]. There have also been increased civilian fatalities since the late 1970s in peri-urban or wildland-urban interface (WUI) areas of the Mediterranean [14]. The WUI is defined as "the area where houses exist at more than 1 housing unit per 40 acres and: 1) wildland vegetation covers more than 50% of the land area or 2) wildland vegetation covers less than 50% of the land area, but a large area (over 1,235 acres) covered with more than 75% wildland vegetation is within 1.5 miles" [15]. This area is important in wildfire management for two reasons: 1) Increased human presence in the WUI will result in more human caused ignitions and thus wildfires, and 2) wildfires occurring in this area pose a greater risk to structures and lives than wildfires in more rural areas, are harder to manage, and must be suppressed [16].

Indeed, these large and severe wildfires in the WUI are becoming more frequent [17] and increasingly impacting different types of communities and human settlements [18]. Population growth, real estate markets, and urban development in some regions might be forcing more people into peri-urban areas of high fire risk. In the US, there are now an estimated 49 million residential homes in the WUI [19]. The increasing number of homes in the WUI has been shown to result in more destructive fires and greater socioeconomic impacts [20]. In California, most structures destroyed from 1992–2015 were not in the forests, but rather in other WUI vegetation and land use types [21]. Elsewhere, approximately 16% of Chile's population now live in the WUI, and wildfires are becoming more frequent in these densely populated areas [22]. The 2019–2020 Australian wildfire season also occurred in more populated areas resulting in a greater number of people affected and greater destruction to property [4,23]. Similarly, the Iberian Peninsula wildfires of 2017 burned primarily in peri-urban residential areas, with a higher incidence inside the WUI in comparison with previous years [24].

Furthermore, different types of human settlements (e.g., urban, WUI) and demographics (e.g., underserved communities, ethnic/racial minorities) are now increasingly at risk of wildfire and its related secondary effects (i.e., mudslides). For example, studies in the US such as those of Wigtil et al. [25] and Davies et al. [26] found that socioeconomically advantaged populations (e.g., White and higher incomes) tended to reside in high wildfire risk locations due to desired natural amenities and high property values. Conversely, these authors note that lower income and minority populations have been largely reported to be concentrated in areas

historically at lower risk, such as in city cores and highly urbanized suburbs. However, recent housing price increases in urban areas of North America [27] increases in peri-urban, less formalized settlements in Latin America, Africa, and Asia [28], and the arrival of immigrants in the Mediterranean region are changing the demographic and socioeconomic make-up of the WUI and communities at risk from wildfire [29]. Amenity migration by higher income populations to peri-urban areas also puts existing residents, often vulnerable populations, at risk through a larger number of homes to defend in case of wildfire, especially since newer residents often choose not to engage in fuel reduction projects [30].

Given the amount of scientific literature on, and media coverage of, wildfires there is a need to understand how, and what types, of communities and people are now being affected by wildfires; both directly (e.g., houses damaged and destroyed) and indirectly (e.g., human well-being, environmental quality, socioeconomic disruptions). Additionally, there is a need to assess the state of knowledge, especially on underserved and disadvantaged communities, and identify research and information gaps.

By providing insights into the timing and scope of the literature on this issue, such studies can help guide future work in this field. This is particularly important to land and environmental managers who are working to incorporate social justice (i.e., EJ) considerations into management practices as well as policy formulation and uptake. Further, such work expands the conversation on wildfire to include the socio-demographic and economic impacts as well. Explaining and quantifying the ways in which community differences are included in the literature allows researchers to better understand how these natural disasters distribute damages disproportionally, not only across landscapes, but also across communities and society. In doing so, practitioners working in conjunction with policy makers can craft more socially equitable policies (e.g., home insurance, community recovery programs) as we will present in this study.

## 1.1. Defining environmental justice

Environmental Justice (EJ) is a broadly encompassing term with numerous nuanced interpretations according to context and the specific environmental problem, and thus there are several widely accepted definitions in the literature [31–33]. Although the term has been used since the 1970s, Bullard's definition [34] is often cited and is, "the principle that all people and communities are entitled to equal protection of environmental and public health laws and regulations". Historically, most environmental justice research has focused on unequal exposure to health hazards (e.g., hazardous waste and air pollution) and concomitant health risks. In 2005, the widespread damage caused by Hurricane Katrina in the US, and the subsequent governmental failures to properly address the disaster, led to emergence of research into the social injustices related to natural disasters [35,36]. In terms of natural disasters, EJ research has investigated if specific disadvantaged populations bear disproportionate impacts from the event [32,37]. Similarly, EJ is also used in the social sciences as a theoretical framework and is known by other terms (i.e., environmentalism of the poor) in different regions of the globe and is now becoming institutionalized and used for policy-making in many countries [38]. For further background and details on the application of EJ in governance processes in the US as related to wildfire, please see text in S1 File.

## 1.2. Aims and objectives

Based on EJ definitions in the literature and policy documents [37,39,40], and for the purposes of this review and to focus our methodology, we propose the following definition of environmental justice specific to wildfires: *when all people, especially those that have not been*

*historically engaged, consulted, and meaningfully involved in governance processes that affect their environment, are inequitably located in high fire risk areas and/or under conditions that make them more susceptible to prolonged exposure to wildfire impacts, smoke or post-fire hazards such as flooding.* Additionally, to quantitatively analyze the disparate socio-demographic dimensions literature, we classified EJ as being characterized by four components:

1. Impacts/Harm: No disproportionate impact from environmental harm on disadvantaged communities or individuals. Examples: Locations where, disadvantaged communities are forced to live in a high-risk fire zone, in an area/home that will suffer from prolonged exposure to fire-related smoke/floods, or live in fire prone areas that have not had recent prescribed burns or fuel treatments;

2. Governance: equal and meaningful access to environmental information and participation in decision making. Examples: transparent access to culturally relevant information on fire-proofing your home, participation in land management and planning processes;

3. Amenities: Equal access to environmental benefits such as clean drinking water, sanitation, and parks. Examples: Access to good quality air, clean water after a fire;

4. Remedies: Access to justice and effective remedies for environmental harm. Examples: Post-fire assistance/aid, access to new/temporary housing post-fire.

Therefore, the aim of this study is to use environmental justice as a lens to better understand the state of the art of the role of wildfire in communities and human settlements. Specifically, this study had three objectives: 1) determine what aspects of the socio-demographic dimensions of wildfire are most frequently studied; 2) identify the geographies and contexts that have been the focus of these studies; and 3) given 1–2, what aspects of environmental justice are being addressed in the socio-demographic dimensions of wildfire literature. We note that different geographies and context will require or have different definitions and relevance; as such here we focus on the English language literature and human settlements principally in the US and in temperate, Mediterranean vegetation types which also occur in Europe, South Africa, Chile, North America, and Australia.

## 2. Materials and methods

### 2.1 Systematic literature review

We used the Preferred Reporting Items for Systematic Reviews and Meta-Analyses (PRISMA) standards [41] to guide our literature review. We first used the Scopus database to search for peer-reviewed scientific articles and reviews written in English matching pre-determined key words and phrases (Table 1). We chose to include technical reports since several land and fire management institutions (e.g., USDA Forest Service) have published extensively in this format. Perspective and commentary type publications were also included; but books, book chapters, and dissertations were excluded. "Wildfire AND health" or "Wildfire AND smoke AND health"; and other similar terms, were not used due to existing reviews on the health effects of wildfire smoke and the need for EJ considerations in the available public health literature (S1 Fig) [42].

The key word searches in the title, abstract and keywords of the publications were carried out in May 2021, returning a total of 1366 publications. After all the searches were completed, we combined the results and used a Quality Assurance and Control (QA/QC) process to remove duplicates (356) as well as book chapters and publications not in English (n = 16). Next, titles were screened for relevance to the research objectives. Selected reports had to meet two criteria: 1) explicit focus on wildfires (including prescribed fire, risk of wildfire, defensible

**Table 1. Search terms used in the review of the socio-demographic dimensions of wildfires literature using an environmental justice lens.**

| Group | Search terms | Justification |
|---|---|---|
| 1 | Environmental justice and wildfire, wildfire and minorities, homeless and wildfire, insurance and wildfire, wildland urban interface AND fire AND justice, wildland urban interface and fire and vulnerability, urban forest and fire and vulnerability, vulnerability AND wildfire AND community | Identify populations known to be most affected by wildfire |
| 2 | Community readiness AND wildfire, Firewise communities, fire resistant home, defensible space, housing AND wildfire | Identify how housing characteristics can influence wildfire effects on people |
| 3 | Urban forest AND fire AND social, wildland urban interface AND fire AND social, wildland urban interface AND fire AND community | Identify relationships between wildfire and communities in urban and peri-urban areas |

space/wildfire mitigation, wildfire smoke, secondary effects of wildfires such as mudslides); and 2) discuss people and/or human communities (e.g., adaptive capacity or the ability to prepare for and recover from wildfires, impacts of wildfires to people/human communities, socio-demographic characteristics such as income and race).

Reports were eliminated at this stage (n = 453) only if there was an obvious mismatch with the above criteria. For example, the concept of "defensible space" is also used in the literature on urban planning and crime. For review of the abstracts and full text, the reports were loosely grouped into subjects and then divided between the multi-disciplinary authors, with each author largely reviewing articles in their area(s) of expertise. Another 359 reports were eliminated after a full reading; leaving 182 studies from Scopus to be included in the review.

Given the common use of more than one database in similar literature reviews, in July 2021 we also searched the Web of Science Core Collection database using the same search terms (Table 1) and protocols. Reports that were new (i.e., had not appeared on the Scopus search, n = 774) were analyzed for inclusion in the review using the same process outlined above. A further 114 studies were added after eliminating those based on title (n = 124), abstract (n = 456), or a full reading (n = 80). This resulted in a total of 296 reports included from the two databases. The third and final source was reports (n = 53) that had referenced Davies et al. [26], a highly cited paper on the environmental justice implications of wildfires. Three studies were identified that matched the above criteria; and were thus included in our final database (n = 299; Fig 1) for subsequent analyses.

## 2.2 Coding and analysis of the literature

All selected publications in our final database were then individually read and coded following a protocol of pre-determined attributes based on our overall literature review and study objectives (S2 File). Specifically, seven attribute categories were developed to characterize the aspects most relevant to our aim. We defined community type according to the location of the study area or socioeconomic group of interest into rural, WUI, urban, or a combination of two or more of these. The time period(s) of the study was classified as at least one of four options: before wildfire/prescribed fire (i.e., plans to evacuate, preparing the home for wildfire), during wildfire/prescribed fire (i.e., evacuation during a fire), post-wildfire (i.e., recovery after a wildfire), and modeled wildfire.

We also coded the studies for what aspect(s) of wildfire (e.g., wildfire, smoke, floods) was explored under hazard, with prescribed burning and mechanical thinning coded under

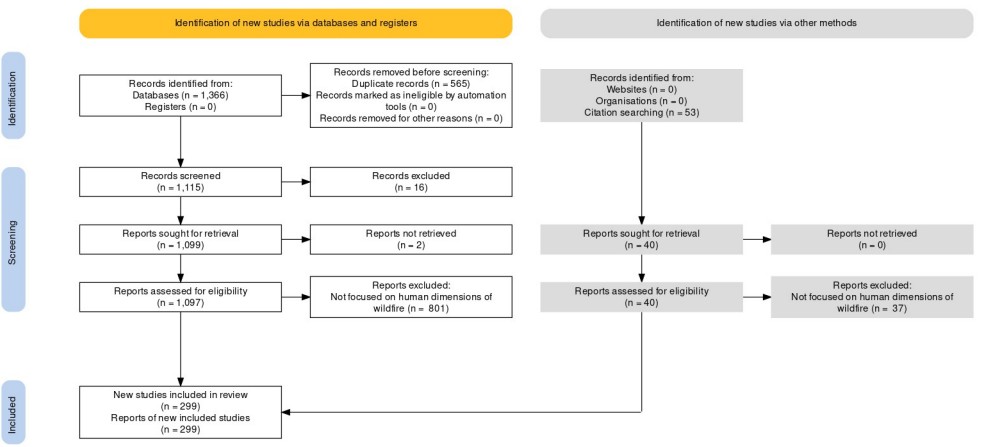

**Fig 1. Diagram of record search and report selection based on PRISMA guidelines.**

"other". For the EJ attribute, we classified the components as explained in section 1.2 to provide a more detailed statistical analysis of our study objectives. Finally, the socio-demographic category was coded to specify different attributes used in the relevant literature (e.g., age, gender, language). Importantly, simply including the attribute(s) when describing the study community/sample population was not sufficient to be coded "yes". Instead, the influence of the attribute needed to be examined, analyzed, or studied to code the study "yes" for the socio-demographic variable.

The attributes were initially coded by research assistants and then two of the researchers conducted a QA/QC review (i.e., full reading of articles and assuring consistencies in applying definitions and coding protocol) of approximately 25% of the studies. At this time the keywords (author's keywords, not the index keywords) underwent a QA/QC process to combine different versions of the same term (e.g., wildfire and bushfire); and author names were also standardized to prevent duplication. We then used bibliometic and statsitical analyses to test our research objectives. For specific details regarding analytical methods, models, variables, and softwares used please see S2 File.

## 3. Results

### 3.1 Performance analysis and sample literature characteristics

In total, 299 studies were selected for analysis: 275 original research articles, 17 reviews, 5 reports, 1 letter, and 1 note. The publications spanned 2002–2021 (Fig 2A), suggesting that the socio-demographic dimensions of wildfire is a relatively new field of study with an annual growth rate of 19%. During 2002–2010 there were fewer than 10 publications each year (Fig 2B). An increase in 2009 appears to mark a turning point in the literature as the yearly production was over 15 articles afterwards (excluding the steep drop in 2010). On average, documents were cited 24.7 times; with later documents being cited more times on average (Fig 2C). Overall collaboration was low as indicated by a collaboration index of 2.65 (i.e., a co-authors per article index calculated only using the multi-authored article set).

### 3.2 Geographical scope, contexts, and collaborations

In terms of the geographical focus of the publications, almost all (97%) discussed only one country, and seven (2%) were modeling and/or review studies not tied to specific countries.

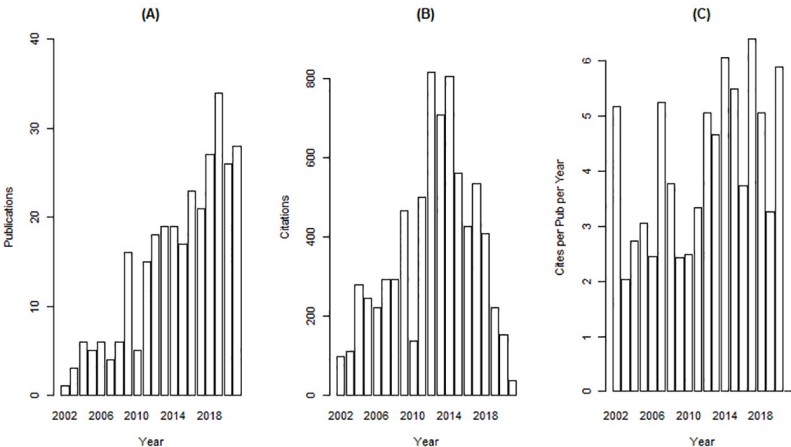

**Fig 2. Number of articles and citations of on the socio-demographic dimensions of wildfire by year.** A) Publications per year, B) Citations per year, C) Citations per publication per year.

More than half (66%) were in the United States (S3 Table in S3 File); and two other countries accounted for another 23% of the studies: 15% in Australia, and 8% in Canada. Spain (2%) and Chile (1%) were the only other countries that were the focus of more than one document. Another measure of international collaboration is multiple country publication, or the number of documents in which there is at least one co-author from another country. By this measure, Spain (0.57), Portugal (0.25), and Australia (0.22) had the highest scores, but Canada (0.083) and the United States (0.077) scored much lower.

In the United States, the western states were the most frequently studied (S3 Fig in S3 File) were California (51 documents), Oregon (48), and Colorado (44; S3 Table in S3 File). The only non-western state in the top ten was Florida (21). Conversely, states in the Northeast were the least represented in the literature. In Canada, the province of Alberta was included in the most publications (12) followed by Ontario (5; S3 Fig in S3 File); and in Australia, the state of Victoria had the highest number (17) closely followed by New South Wales (13; S3 Fig and Tables in S3 File). S3 Fig in S3 File provide the geographical focus for the US, Canada and Australia and the collaboration networks among different countries, respectively.

A co-occurrence network of keywords showed seven clusters and 174 links (Fig 3). There were 763 author provided key words, with "wildfire" and "wildland-urban interface" occuring the most often, and having links with all clusters. Cluster 1 (red) was the largest, focusing on risk and the wildland-urban interface. Relevant to the aims of this study were clusters 5 (blue) which contains EJ as a keyword, and cluster 4 (green) which focuses on prescribed fire; but also includes literature around social vulnerability to wildfire.

We also used the author keywords to plot the document themes into four quadrants depending on centrality and density rank values (For specific methological details, please refer to S2 File). Fig 4 shows that the upper-right quadrant ("motor themes") displays well developed, well-established, and important themes in this field of research (e.g., defensible space, wildfire mitigation, and attitudes). The upper-left quadrant displays very specialized (i.e., "niche") and terms with marginal importance to the field of research (collaborative planning). Themes in the lower-left quadrant are "weakly developed and marginal", indicating they are either "emerging or declining" themes. There were two clusters in this quadrant, one around risk (e.g., mitigation, adaptive capacity, and risk assessment) and another containing keywords such as community, communication, and alternatives to evacuation. However, the lower-right

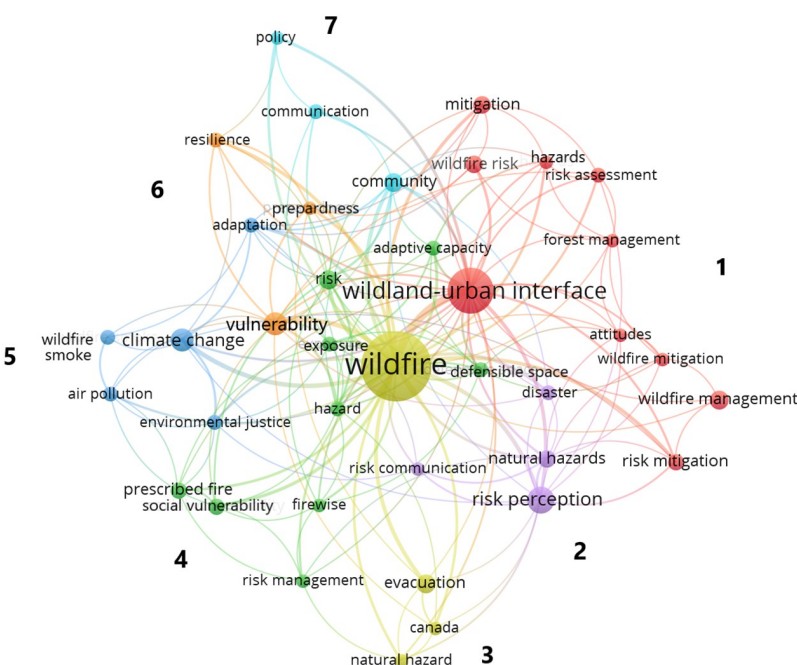

**Fig 3. Network of author-identified keywords in the socio-demographic dimensions of wildfire literature with a minimum occurrence of five.** Total link strength = 394. Numbers refer to the different clusters.

quadrant displays important but not yet fully developed "basic" themes. Included in this quadrant are EJ and vulnerability, highlighting scant research and the emergence of these themes in the literature.

## 3.3 Environmental justice in the socio-demographic dimensions—Wildfire literature

A substantial proportion of publications (29%) in our collection did not focus on a specific community type or study area, and thus included a range of communities along the urban to rural gradient (S3 Table in S3 File). The largest proportion (44%) of publications focused solely on communities located in the WUI; while communities that were either urban (2%), or urban and WUI were the least studied (0.3%). The majority (68%) of publications focused on the temporal period before wildfire occurrence, such as defensible space; while both during (e.g., evacuations) and after (e.g., rebuilding) wildfire events were the focus of less than one fourth of the publications (20% and 29%, respectively). Age (56%) and income (54%) were included in more than 50% of these publications, but language (11%) and transportation (8%) were rarely discussed. The most common "other" attribute was residency (i.e., full-time or seasonal resident of a community; S3 Table in S3 File).

In terms of EJ, 20 percent (60) of all reviewed publications met our definition of having addressed issues of environmental justice. The first EJ publication was in 2004, but EJ was first specifically identified as a topic in 2018 (Fig 5); and approximately 50% of the EJ literature was published after 2019. Most of the EJ literature (80%, Fig 6) focused on impacts and/or harm, and governance/information was the second most discussed aspect (51%). There was also a notable number (7) of publications that included all four components of EJ.

The EJ relevant publications were from 11 countries, along with two regional and a global scale study. The United States had the most (30), followed by Canada (11), Australia (7), and

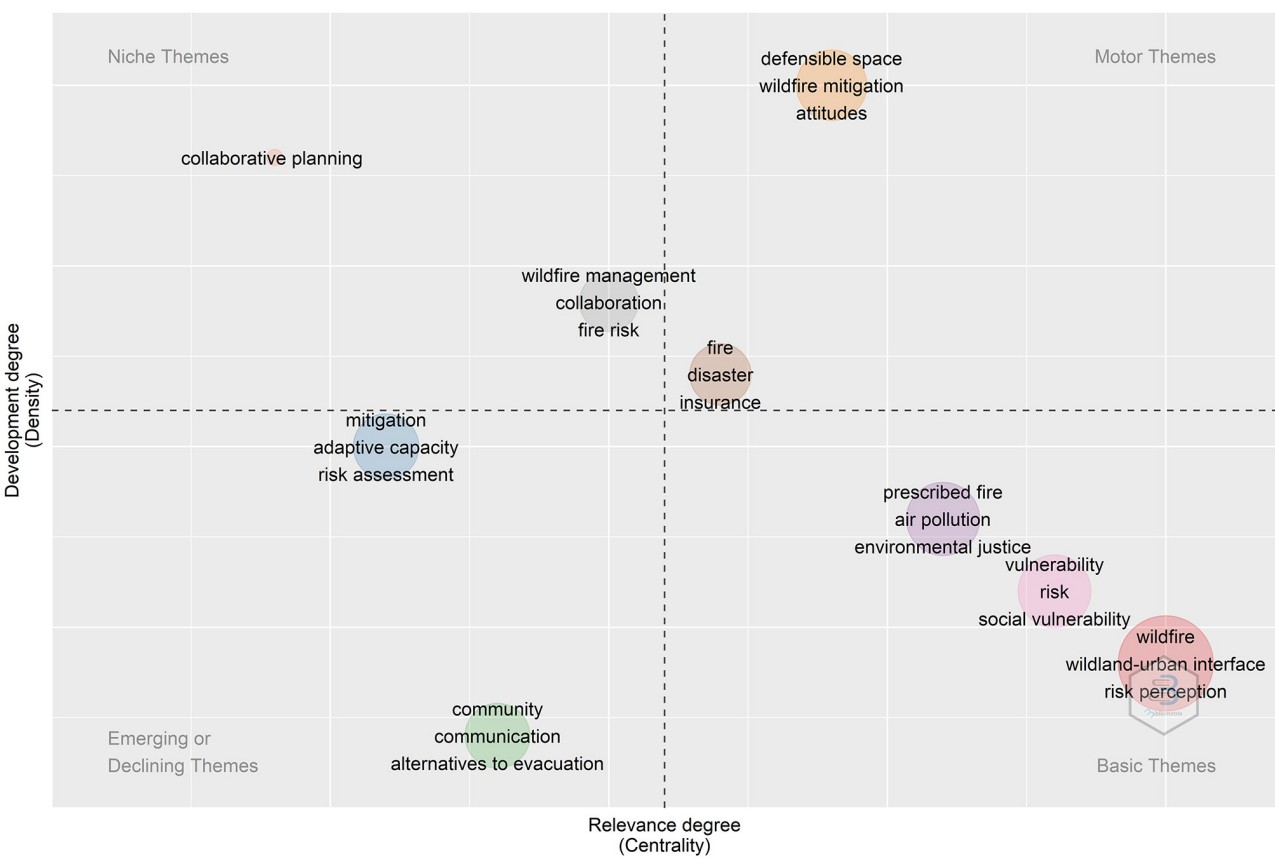

**Fig 4. Thematic map of development and relevance degree of the socio-demographic dimensions of wildfire literature based on author provided keywords.**

Chile (3). Algeria, Argentina, Italy, Ivory Coast, Mexico, Portugal, and Spain each had one EJ publication. However, when measured by the percentage of a country's total publications, Chile had the highest percentage (75%) out of the four countries with multiple EJ publications; followed by Canada (44%), and Australia and the United States (16% each). Four of the other

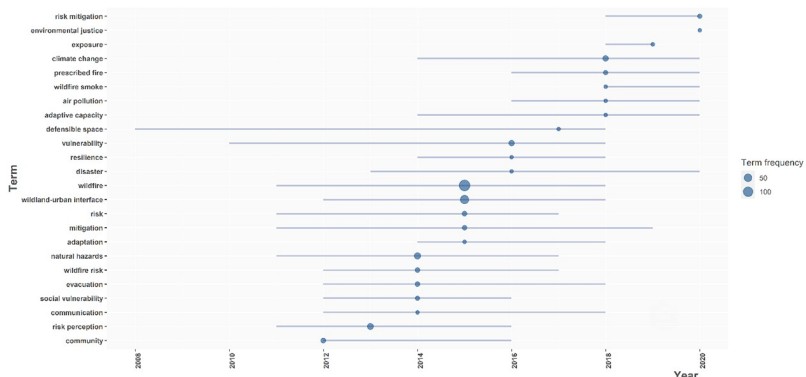

**Fig 5. Temporal trends in topics and the frequency of the use of that author keyword in the socio-demographic dimensions and wildfire literature.**

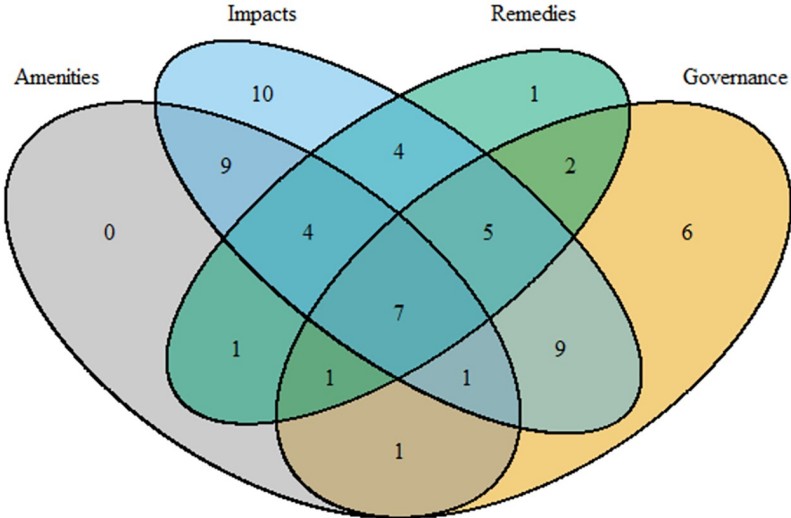

**Fig 6. Venn diagram of number of papers for each of the four components in our definition of environmental justice.**

countries (Argentina, Algeria, Ivory Coast, Mexico) had their only publication in the collection classified as EJ. The proportions for the remaining countries were as follows: Italy (50%), Portugal (25%), and Spain (14%).

Our probit model statistically confirmed that there are more EJ designated papers in recent years, indicating a positive trend in the number of EJ papers over time (Table 2). The WUI and rural—WUI designated community types decreased the likelihood that a study was EJ relevant. Although the type of hazard studied was not significant, papers examining the post-wildfire period were more likely to be EJ than those considering another time period. In terms

**Table 2. Results of the probit model for variables significantly related to the classification of a publication as relevant to our definition of environmental justice (n = 299).**

| Variable | Binary dependent variable: Environmental Justice Designation | |
|---|---|---|
| Year | 0.047* | (-0.028) |
| Community: Rural and WUI | -1.376** | (-0.615) |
| Community: WUI | -0.505* | (-0.279) |
| Race | 0.940** | (-0.38) |
| Poverty | 2.126*** | (-0.529) |
| Period: post | 0.531* | (-0.273) |
| Region: USA | -0.723*** | (-0.28) |
| Constant | -95.649* | (-56.064) |
| Log Likelihood | -90.253 | |
| Akaike Info Criteria | 228.505 | |

* $p<0.10$;

** $p<0.05$;

*** $p<0.01$.

Note: Standard errors are in parentheses.

of socio-demographic characteristics, only race/ethnicity and poverty significantly increased the likelihood that a study was EJ (Table 2). Chi-square tests of the relationship between EJ and socio-demographic variables also found that both race/ethnicity and poverty were highly significant (both $p < 0.01$). Additionally, employment and gender (both $p < 0.05$), and language ($p < 0.10$) were also significant. In addition to the regression results, we also provide an analysis of deviance table for our probit model (See S1 Table).

## 4. Discussion

Globally, recent record-breaking wildfires and wildfire seasons have highlighted the need for a greater understanding of the socio-demographic aspects of these events, especially given their often-unprecedented nature and emerging impacts to society. By proposing an environmental justice-wildfire relevant definition, we analyzed how this emerging concept is gaining interest across research in the natural resources and environmental sciences. Below, we discuss our findings relative to our three research objectives and present areas of future research before concluding.

Our first objective was to determine what socio-demographic aspects of wildfire were most frequently studied. We found that studies of the socio-demographic dimensions of wildfires is a relatively new field, as shown by 2002 being the first year of publication for a document (Fig 2). This is consistent with McCaffrey [43] who noted there were few publications studying wildfire as a natural hazard, as those types of studies instead covered hurricanes, floods, and earthquakes. Notably, the number of publications began substantially increasing in 2009. This increased interest may be reflective of a shift from wildfires in predominantly rural and sparsely population areas to more populated peri-urban and urban areas. Examples of wildfire events occurring around 2009 that reflect this change include the Black Saturday wildfires in Australia that killed 173 people and destroyed more than 2000 homes [23]; and the Station Fire in Los Angeles, US, the largest wildfire in the modern history of Los Angeles County which threatened 12,000 homes and structures [44].

Davies et al. [26] note that their study on wildfire vulnerability mostly excluded urban areas. However, as people and poverty move from city cores that have low wildfire risk to suburbs and communities in, or adjacent to, the WUI [45]; their vulnerability to wildfire is likely to also change. Despite this demographic trend, little is known on how minority populations are being impacted by wildfire as the demographics of the WUI change [29]. For example, African American US populations are currently largely located outside of areas at high risk of wildfire. However, they have high social vulnerability/low adaptive capacity (lower incomes and education, lack of transportation) so would respond poorly in the event of a wildfire [26].

Similarly, certain socio-demographic groups are more vulnerable to wildfire smoke effects due to factors such as pre-existing health conditions, and a lack of insurance/medical care. In Australia, effects of wildfire smoke on health tended to be stronger for Aboriginal populations [46]. However, in their review of the physical health effects of non-occupational wildfire smoke, Liu et al. [47] found that only three out of 61 (5%) studies reported "effect modification by socio-economic status, race, or co-morbidities" (p.128). Similarly, Reid et al. [48] reviewed 115 articles on the health impacts of wildfire smoke exposure and noted that "Few epidemiological studies have investigated whether specific populations are more susceptible to wildfire smoke exposure than the general population" (p.1340).

Indeed, smoke, either from wildfires or prescribed burns was the second most common hazard studied, despite not specifically including smoke and health keywords in the search. This is perhaps due to growing awareness of its impact on distant and larger metropolitan communities [19,49]. Wildfires in the Western US are increasingly causing ambient air quality to exceed

standards in distant regions of the country; leading to significant health complications including increased risk of respiratory disease, cardiovascular disease, and mortality [42,47].

Despite the increased occurrence of wildfires in more populated areas, our literature review identified few explicit studies that analyzed or assessed how wildfires are affecting "homes" and people and communities—according to their socio-demographics—across space and time. Such information using available geospatial data or socio-ecological frameworks could provide decision-makers with improved information and a better understanding whether indeed different groups (e.g., disadvantaged or urban communities) are now being increasingly impacted by wildfire in recent years as reported in the media. For example, in the US, climate change is projected to increase the level of risk in most regions. Masri et al. [50] found that in California US, the elderly and low income were disproportionally affected by recent wildfires. Thus, wildfire risk is also likely to increase across different socio-demographic groups by the urban-rural migration that increasingly brings new demographics to the WUI [27].

Our second objective was to identify the geographies and contexts that have been the focus of previous wildfire-related studies. Geographically, most of the publications were from English speaking and high-income countries (S3 Fig in S3 File); with Australia, Canada, and the US still comprising the majority. In part, this is due to our analysis of English language-based publications and the size of these countries; but also, the large costs (i.e., destroyed homes and fatalities) of their wildfires [51]. Asia was poorly represented, with only one study (Thailand) [52]; and there were only two African countries (Algeria and Ivory Coast) in our analyzed literature. In Europe, research was also unevenly distributed; most were from Mediterranean countries such as Spain and France, although Greece had one publication [53].

In the US, almost 78% of all houses at high risk of wildfire and with high social vulnerability were in just seven states: California, North Carolina, Florida, South Carolina, Georgia, New Jersey, and New York [25]. Despite this, only two states (California and Florida) were in the top 10 for the number of US based publications that focused on individual states (Fig 3A; S3 Fig in S3 File). Indeed, these costly and highly severe wildfire events are reactively and likely resulting in greater subsequent scientific productivity in certain regions and countries on these topics particularly related to response; as opposed to proactively investing and studying planning and response related issues [54].

In terms of community types and context, the WUI was most studied (50% of community types; Fig 3 and S3 Table in S3 File). Research focusing on rural and urban communities was scarce, with the former likely reflecting the smaller population directly affected, and the latter the relatively recent incursions of wildfires into these areas. We found that US publications often emphasized social acceptance of techniques to mitigate wildfire risk (e.g., fuels reduction on public and private lands) and more recently, public response during and after wildfires. Conversely, in places (e.g., Australia) where many people might "shelter in place", instead of evacuating, risk perception, homeowner preparedness, response during wildfires, and community safety were key areas of research. These findings largely confirm other studies that indicated that the pre-wildfire period has been the most studied time period [55]. In terms of post-fire impacts, only 3% of the articles in this review studied post-wildfire effects such as mudslides; presenting an opportunity to assess the role of wildfire on non- air quality related ecosystem disservices (i.e., costs) with a focus on their supply, demand, and subsequent effects on human well-being. For example, Wan et al. [56] researched the effects of ash from wildfires on soil quality and the associated risks to farmworkers. Another study [57] notes that natural hazards research usually focuses on a single hazard rather than multiple connecting hazards (e.g., wildfire and subsequent flooding).

As climate change impacts increase throughout the world, more communities—both plant/fuels and human—will be affected by extreme events such as wildfires. However, these effects

are unequal as the most vulnerable communities (i.e., EJ) usually suffer from the strongest impacts; and many of these communities do not have the resources to mitigate, recover, or adapt from these effects. Therefore, policy makers need to identify vulnerable communities and the barriers they face to better understand how to develop strategies to help reduce impacts from increased wildfire occurrence and severity.

Accordingly, our third objective was to better understand what aspects of environmental justice are being addressed wildfire literature. We found that most fuel treatment literature understandably focused on WUI areas, but that the studies that focused on WUI areas had a lower likelihood of being labeled as EJ-relevant (S3 Table in S3 File). Our review also showed that studies focused on the pre-fire period, including a large number of defensible space studies, have a lower likelihood of being labeled as EJ (Table 2).

Overall, studies from our review that focused on EJ described different impacts to, and the social vulnerability of, different communities. For example, Masri et al. [50] found that except for Native American communities, wildfires impact US census tracts with lower populations of minorities and higher populations of elderly. Elderly populations are often dependent on external assistance for evacuation [50] and are also highly susceptible to smoke-related health conditions [47]. Furthermore, the elderly population is more likely to have fixed incomes, making them more likely to relocate to peri-urban zones due to housing price increases [45], and have limited capacity to address the financial hardships of a wildfire. In addition to being disproportionately located in areas of high fire risk, Native American communities also have high social vulnerability and low adaptive capacity [26]; including high levels of poverty, low levels of education, and commonly experience household crowding.

Our review found that socio-demographic, EJ, and wildfire relevant studies significantly focused on race/ethnicity and poverty aspects, and to a lesser degree on employment, gender, and language (S3 Table in S3 File). Indeed, race, class, and poverty have often been at the core of traditional EJ studies [32,34,45]. However, Sotolongo et al. [58] note that "most literature on disaster recovery that incorporates demographic and/or socioeconomic factors does not explicitly invoke environmental justice analysis techniques. . ." and that race/ethnicity and poverty are often insufficient in making a study EJ-relevant (p.61). Thus, our review confirms that simply including socio-demographic variables (e.g., age, income, race) in a methodology or analysis without specifically analyzing or addressing a historically or traditionally disenfranchised community's issues, is not sufficient to define a wildfire study as having EJ implications.

Despite the literature's focus on pre-wildfire, studies were more likely to consider EJ topics when focusing on the post-wildfire period. Villagra and Paula [59] also show that the post-wildfire effects research and its role in human well-being literature is generally not specific to after the wildfire; but instead, often covers this time in addition to before, or during, a wildfire. We also found that both social vulnerability and environmental justice were not yet fully developed "basic" themes, highlighting the scant research and future research potential of these topics (Fig 4).

It is also notable that the four components of EJ in our proposed definition were not equally studied in the literature, with impacts/harm researched more frequently than the other three components (Fig 6). This finding is consistent with other natural hazards and disaster research in that there is less focus on remedies (i.e., aid distribution post-disaster; [60]). Future EJ studies could also be broadened to include more research on aspects aside from the impacts, such as governance. Participation in governance processes is an important aspect of EJ; as the capacity to participate in governance can directly influence the outcomes [61]. However, preparation for wildfire has been shown to affect outcomes, and access to preventative measures may not be equally distributed across different communities and societies equitably [54].

Regarding language, we found scant research documenting how mastering the predominant language can compromise an individual's ability to act during or after a wildfire (e.g., English in the US, Spanish in Spain), and can make it more difficult for certain migrant groups to respond to wildfires and access health care, housing, and other social services post-wildfire. Our bibliometric analysis of EJ coded publications also found that many studies did not specifically use "EJ"; but rather other terms, including social justice, inequities, and social vulnerability (Fig 5). Thus, understanding EJ as applied in our definition, can help guide future research and policies around the socio-demographics of wildfire effects, and its changing nature.

Although our study presents several important findings, we were also bound by some limitations. First, we note that our review methods omitted some relevant global literature since it was limited to English language publications indexed in available databases. This means that that many relevant publications from frequently wildfire affected Mediterranean and Latin American regions were likely excluded. Second, our search terms greatly delimited the bounds of the studies considered, since for example, "wildfire evacuation" and "wildfire smoke" were not used as search terms.

That said, our study is one of the first to review the socio-demographic dimensions and environmental justice implications of the wildfire literature. Accordingly, we identify several topics, contexts, and topical areas in need of further research. This is particularly relevant since future climate and global change projections indicate an increase in wildfire risk in many global regions; even those not historically at high risk [54]. These increasingly common wildfire related impacts have increased concerns over the potentially inequitable impacts on human settlements, particularly vulnerable and disadvantaged communities.

For example, although Fig 4 shows that prescribed fire is still considered a basic theme, there is still work needed in this field. In particular, incorporating socio-demographic and EJ considerations into fuel treatment studies holds considerable promise for better understanding issues of equitability and inclusivity. The results of our review also suggest that a more interdisciplinary research approach to wildfires might be needed to address this growing crisis, including not only wildfire and subsequent smoke effects, but other potential secondary effects such as flash floods and impacts to livelihoods.

Given the emergence of environmental problems and changing demographics in peri-urban areas in landscapes at risk of wildfire, socio-economic attributes such as: age, education, governance, language, housing status (i.e., rentals and multi-family housing) are key factors that could be addressed in the wildfire -socio-demographic dimensions-EJ literature [50,62]. Thus, there is an opportunity to assess current and future socio-demographic trends within the WUI to establish which communities are most at risk from wildfire and its effects. Carrol and Paveglio [63] highlight that community characteristics are important for investments in fire mitigation behavior, and that different socio-demographic groups may require different approaches for community level wildfire mitigation. Fig 3 shows the connection between communication and community preparedness, demonstrating that information is important for home mitigation. Thus, as corroborated by our literature review, understanding socio-demographic characteristics is important for communicating risks more effectively [64].

Overall, our literature review indicates that different groups of people are now being affected by wildfire regardless of income, race, and education. Similarly, historically disenfranchised people and newer more diverse groups living in peri-urban, or more urban settings, might be increasingly vulnerable due to housing issues, adaptive capacity, and the increasing impacts of large, severe WUI fires and subsequent smoke emissions. Therefore, our study highlights the potential of future research to focus on environmental justice issues related to the preparation for large wildfire events across the globe.

More research on the topics identified in this study could help regional and local governments better mitigate wildfire effects on diverse, less powerful, and less influential communities and improve their resilience. For example, another area for future research is understanding wildfire effects on indigenous communities [33,65]. Our review found that most publications that focused on indigenous communities were from Canada [66]. In the United States, although there is substantial literature on Native Americans, it often focuses on prescribed fire planning and cultural burning [67]. Furthermore, none of the Australian studies included in this review focused on Aboriginal and/or Torres Strait Islanders; despite them being "disproportionately" affected by the 2019–2020 wildfires [68]. Thus, further research is needed on the effects of wildfire on Native American and indigenous communities; especially considering their high social vulnerability.

Post-wildfire recovery and trajectories is also another area that would benefit from increased research. For example, tracking wildfire displaced persons due to evacuations and home loss, and their ability to adapt, would increase our understanding of how different communities, particularly disadvantaged ones, understand and respond to these events and ultimately the factors that influence the recovery trajectory. New and uncommonly used data sources such as georeferenced cell phone data, "big data", economic activity, and construction permits could be utilized for such studies [10].

Wildfire now poses a range of threats to livelihoods, life, and health across all rural, peri-urban, and urban areas. This possibly indicates that the fuels and vegetation-based definition of WUI as applied in fire management might be more applicable to rural contexts and communities. Thus, future research could also develop more region-specific definitions, socio-ecological criteria, and metrics for use in rapidly growing and more highly populated and complex peri-urban areas not only in the US, but elsewhere [69]. As urbanization alters peri-urban areas and urbanized communities and their infrastructure becomes increasingly impacted by wildfire, there will be a need to incorporate urban ecology and geography related research frameworks [31,45]. Our findings also confirm the need for continued research not only in higher income countries frequently affected by wildfires, but also in other regions in medium and low-income countries with increasing occurrences of wildfires, especially in peri-urban areas, that may not receive as much international attention [54].

Given the above, one of the key areas for research will be understanding the increasing effects of wildfire on peri-urban socio-ecological systems and their structure, function, resilience, and ecosystem services and disservices [23,28,70]. This research will be especially needed in terms of applicability to historically disadvantaged communities and their language, cultural, and vulnerability contexts, as well as their resource, housing, and educational limitations. Further research is warranted into how wildfire and associated effects impact agriculture and other natural resource-based livelihoods; as well as employment-related opportunities, displacement, and relocations associated with post-fire construction and restoration activities.

## 5. Conclusion

Given the emergence of wildfires as one component of global climate change, socio-ecological communities need to adapt and become more resilient to wildfire. Wildfires are like other natural disasters and could be analyzed using a socio-ecological, rather than strictly through a land, resource, or fire management lens. Hence, researchers can begin to delve more deeply into the socio-demographic aspects of wildfires and explore how disadvantaged communities are currently being—and will be affected in the future—by larger and/or more severe fires occurring in more highly populated and diverse (and often vulnerable), communities. Indeed,

most of the studies we identified were from high income North America and European counties as well as Australia.

In conclusion, it is our hope that our proposed definition of EJ can be used by both governmental and non-governmental organizations to guide the inclusion of EJ in their fire, land, and environmental policies and management efforts. However, care is warranted in that EJ should ideally not be used as a quantitative or binary metric. Rather, EJ can be better used as a lens or framework to understand the impacts of emerging environmental problems affecting diverse and historically disadvantaged communities.

## Supporting information

**S1 Fig. A dendrogram developed using Bibliometrix and author provided keywords showing the topic distribution of publications (n = 299) included in the literature review.** Topics associated with wildfire smoke and air pollution in the blue branch, and subsequent effects, are a distinct branch that is separate from the wildfire and socio-demographic topics (shown in the red branches) addressed in this study.
(DOCX)

**S1 Table. Analysis of deviance table for the probit model for variables significantly related to the classification of a publication as relevant to our definition of Environmental Justice.** Note: Df, Degrees of Freedom; Resid., Residual; Pr, Probability.
(DOCX)

**S1 File. Background of environmental justice based on policies from the United States.**
(DOCX)

**S2 File. Table and detailed methodological information on analytical methods, models, variables, and software.**
(DOCX)

**S3 File. Additional results including text, figures, and tables.**
(DOCX)

## Acknowledgments

The authors want to thank Yuli Tovar and Annette Buenrostro for carrying out the attribute coding. We also thank Phil Rodbell, USDA Forest Service, for helpful suggestions. Lastly, we would like to thank the editor and anonymous reviewers for helpful comments that improved the manuscript.

## Author Contributions

**Conceptualization:** Alyssa S. Thomas, Francisco J. Escobedo, Matthew R. Sloggy, José J. Sánchez.

**Data curation:** Francisco J. Escobedo.

**Formal analysis:** Alyssa S. Thomas, Francisco J. Escobedo, Matthew R. Sloggy.

**Investigation:** Alyssa S. Thomas, Francisco J. Escobedo, José J. Sánchez.

**Methodology:** Alyssa S. Thomas, Francisco J. Escobedo, Matthew R. Sloggy.

**Project administration:** Francisco J. Escobedo.

**Supervision:** Francisco J. Escobedo.

**Validation:** José J. Sánchez.

**Writing – original draft:** Alyssa S. Thomas, Francisco J. Escobedo, Matthew R. Sloggy, José J. Sánchez.

**Writing – review & editing:** Alyssa S. Thomas, Francisco J. Escobedo.

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
