## [Decision Letter · Decision Letter 0]

15 Feb 2022

PONE-D-22-00418A burning issue: Systematically reviewing the human dimensions and environmental justice aspects of the wildfire literaturePLOS ONE

Dear Dr. Escobedo,

Thank you for submitting your manuscript to PLOS ONE. After careful consideration, we feel that it has merit but does not fully meet PLOS ONE’s publication criteria as it currently stands. Therefore, we invite you to submit a revised version of the manuscript that addresses the points raised during the review process.

 Please pay special attention to addressing these reviewer comments: (1) explain the importance of this topic; (2) address health impacts; (3) summarize the main findings from the literature; and (4) accurately identify and define the type of review this paper conducts.

We look forward to receiving your revised manuscript.

Kind regards,

Julia A. Jones

Academic Editor

PLOS ONE

Journal Requirements:

2. We note that Figure 4 in your submission contain map images which may be copyrighted. All PLOS content is published under the Creative Commons Attribution License (CC BY 4.0), which means that the manuscript, images, and Supporting Information files will be freely available online, and any third party is permitted to access, download, copy, distribute, and use these materials in any way, even commercially, with proper attribution. For these reasons, we cannot publish previously copyrighted maps or satellite images created using proprietary data, such as Google software (Google Maps, Street View, and Earth). For more information, see our copyright guidelines: http://journals.plos.org/plosone/s/licenses-and-copyright.

a) You may seek permission from the original copyright holder of Figure(s) [#] to publish the content specifically under the CC BY 4.0 license.  

Additional Editor Comments:

All four reviewers saw potential in this paper, but also expressed concerns. Please address the following issues in your revision: (1) explain why this topic is important and to what audiences, (2) consider including the health impacts of proximity to both wildland fire and wildland fire smoke, which is a key element of justice, (3) summarize major findings from this literature and what has been learned, (4) accurately select and define the type of review that this paper contains.

Reviewers' comments:

Reviewer's Responses to Questions

**Comments to the Author**

1. Is the manuscript technically sound, and do the data support the conclusions?

Reviewer #1: Partly

Reviewer #2: Yes

Reviewer #3: Yes

Reviewer #4: No

2. Has the statistical analysis been performed appropriately and rigorously? 

Reviewer #1: I Don't Know

Reviewer #2: Yes

Reviewer #3: I Don't Know

Reviewer #4: I Don't Know

3. Have the authors made all data underlying the findings in their manuscript fully available?

Reviewer #1: Yes

Reviewer #2: Yes

Reviewer #3: Yes

Reviewer #4: No

4. Is the manuscript presented in an intelligible fashion and written in standard English?

Reviewer #1: Yes

Reviewer #2: Yes

Reviewer #3: Yes

Reviewer #4: No

5. Review Comments to the Author

Reviewer #1: This paper is really nicely done from a professional standpoint - rigorous analysis, nicely written, etc. I had only some minor suggestions and then a couple of larger concerns:

Lines 180-3 – probably would be wise to not use the word “injustice” in the definition of justice.

Lines 498-500 – this seems almost tautological.

Line 513-15 – I’m not sure the findings support this claim, simply identifying a gap or an imbalance doesn’t prove the need to fill that gap or remedy that imbalance. There is at least one missing premise in the argument here – maybe about some harm that this gap or imbalance creates?

Line 517 – this is an unnecessary speculation.

Lines 517-19 – while I agree that there are benefits to such collaborations, the argument for them needs to be made independently – I’m not sure the speculation about lack of co-authors is enough to support this argument.

I appreciate the suggestions for future research.

The analysis would benefit by reference to some parallels. I’m not sure what to make of X% of some body of literature mentioning EJ. Is that high, low, average? Could this be compared to some other body of literature? Maybe others have done something similar in other fields, e.g., wildlife.

I think the paper would benefit from an attempt to explain to the reader why this matters. Why does it matter that EJ is discussed in some percentage of articles in a given literature, or maybe why does it matter that EJ is discussed at all? An answer to this question, even a modest one, would make the paper much more engaging.

Reviewer #2: This manuscript presents a systematic review of the EJ literature related to wildfire. The new, wildfire-specific definition of EJ presented was effective and should be shared for more broad use across the interdisciplinary field.

This manuscript addresses three stated objectives to 1) determine what aspects of the human dimensions of wildfire are most frequently studied, 2) identify the geographies and contexts that have been the focus of these studies, and 3) given 1-2, what aspects of environmental justice are being addressed in the human dimensions of wildfire literature. While I agree that these objectives were all addressed, I was surprised at the lack of information that was presented regarding the health impacts of proximity to both wildland fire and wildland fire smoke. This gap was made clear in the fact that none of the top ten cited journals were in health fields. As a growing public health concern, I was surprised by the lack of public health data presented. I think this manuscript is complete as is. However, I think it would add to the discussion if this were addressed as an indication that wildland fire research and public health research need to be brought together to address this growing interdisciplinary crisis.

Reviewer #3: PLOS 1 Review: 1 February 2022

This topic is timely, and the title is intriguing. My first impression is that the MS would discuss substantive findings from the relevant literature. I was hoping to learn more about how EJ factors affect responses to wildfire/prescribed burns. Rather, the MS analyzes components of that framework in detail. This is okay from a technical or descriptive standpoint, but the piece would be more inviting if it summarized major findings from this literature—if it told the reader how environmental justice indicators like race or income or language made for unjust situations. What have we learned so far from this nascent literature? This is what’s really important. This could be summarized by country, region of country, or some other way. I think you can keep all you’ve done but add to it summaries of findings from this literature.

Page 6, lines 106-108. This isn’t always the case. Poor people do live in fire prone areas. I don’t know whether you reviewed Timothy Collins’ papers from the mid-2000s which discuss poor people made more vulnerable to wildfire when rich people moving to ecologically fragile places in the WUI:

Collins, T.W., 2005. Households, forests, and fire hazard vulnerability in the American west: a case study of a California community. Environmental Hazards 6, 23–37.

Collins, T.W., 2008a. The political ecology of hazard vulnerability: marginalization, facilitation and the production of differential risk to urban wildfires in Arizona's White Mountains. Journal of Political Ecology 15, 21–43.

Collins, T., 2008. What influences hazard mitigation? Household decision making about wildfire risks in Arizona's White Mountains. The Professional Geographer 60, 508–526.

There are also two papers from the southeastern US that analyze the intersection of social vulnerability and wildfire risk: Poudyal et al, 2012 (Environmental Management) and Gaither et al, 2011 (Forest Policy & Economics). Gaither et al 2020 also considered exposure to prescribed burn smoke as an injustice—as opposed to Adams & Charnley who looked at the absence of prescribed burning as the injustice.

Your governance/information is similar to existing EJ definitions that describe it as including ‘procedural or participatory’ elements. See Setha Low 2012 for a full discussion. Public space and diversity: Distributive, procedural and interactional justice for parks. In G. Young, & D. Stevenson (Eds.), The Ashgate research companion to planning and culture (pp. 295–310). I think you should acknowledge this and literature that informed the components you include in your definition.

Your EJ definition on p 9 comes off as somewhat top down and authoritative. It dictates where people can live without acknowledging that people are sometimes willing to assume risk. The definition should be reworded in positive terms, that is, in a way that says what EJ is and less what it is not. It should not suggest that people don’t have a choice in where to live.

Page 30, lines 642-643: not sure what this means: “Thus, future research on the socio-ecological dimensions of wildfire would benefit from focusing on human population-based variables and their dynamics as well….”

Reviewer #4: Review comments on PONE-D-22-00418

This manuscripts describes a review of literature on what the authors call the human dimensions of wildfire. It is intended to be a survey of the content and geographic focus of the literature, as well as the extent to which environmental justice as a framework has been applied to this research. Unfortunately, I found the scope and research questions to be very unclear, the methods very unclear, and the results and discussion to lack clear (or useful) framework or message.

A main critique is that this study should not be described as a systematic review. The study aims and questions are very general (and somewhat vague), and the goal is not to compile and compare studies using a specific study design, experimental treatment or technique. Cochrane defines systematic reviews as: “A systematic review attempts to identify, appraise and synthesize all the empirical evidence that meets pre-specified eligibility criteria to answer a specific research question. Researchers conducting systematic reviews use explicit, systematic methods that are selected with a view aimed at minimizing bias, to produce more reliable findings to inform decision making.” I suggest that the authors review the wide selection of writing about different review types and select a more appropriate review definition. Scoping review might be a good selection.

Other specific concerns are listed below.

The authors use a lot of jargon coming from forestry (I believe). For example, while the authors spell out the acronym WUI, they never define it. What is the WUI and why is it an area of relevance in relation to wildfire ecological and social impacts? Same for “peri-urban”

On line 85 the authors switch the focus to WUI, while it is not clear why.

I’m curious, if the authors wanted to include literature on the human health impacts of wildfires and smoke from wildfires, why they did not include search term “health”?

The authors use the term “environmental justice first on line 113, but without defining the term “As such, the increasing vulnerability of peri-urban and WUI areas, and newer socio-demographic groups, to wildfires indicates a need for environmental justice research in this area.”

The 2nd use of the word is in the statement of study aims: “Therefore, the aim of this study is to use environmental justice as a lens to better understand the state of the art of the role of wildfire in communities and human settlements.” This is a big surprise because the authors have not yet defined the term or how it could be useful as a lens.

1.3 Aims and objectives

In this section, I do not find any aims or objectives stated.

Methods

The selection criteria is extremely vague:

“Selected reports that were not filtered out had to meet two criteria: 1) explicit focus on wildfires (including prescribed burns, risk, and defensible space); and 2) discuss the human dimensions aspect (e.g., adaptive capacity, sociodemographic variables).”

What is “the human dimension aspect”?

What is adaptive capacity? Variables are elements of statistical analyses, not topics of study.

This is a critical flaw of the study. There is no way that any other person/group could reproduce this study, which is the main criteria of a systematic review (and any review).

The selection criteria should be explicitly and clearly defined in the main manuscript, not put in an appendix. If the authors need to make space, the selection criteria is far more important than list of journals presented in Table 2.

The authors state nowhere which databases they searched. Therefore I cannot tell what biases might have been introduced in the search, and there is no way that someone could reproduce the author’s findings.

Results

Section 3.1

Which studies and how many were included in the analysis?

On line 217 is the statement “Another 359 reports were eliminated after a full reading; leaving 182 studies to be included in the systematic review.”

But in the Results it states “In total, 299 studies were selected for analysis”

In the References list, there are only about 80 references listed.

This is very unclear.

It’s not clear why the journals in which selected papers are published are listed. Which objective or research question does this information answer/pertain to?

Table 2. I am very surprised that journals such as Environmental Health Perspectives, Environmental Health, Environmental Research are not included in this list. Each journal contains far more than 6 articles on health-related impacts of wildfire.

It is difficult to understand the need, and use for the environmental justice question and results. The authors have described the aspects of environmental justice that are addressed in human impacts of wildfire literature, but give no information about content, which would be more useful. A more useful question might be: what aspects of human impacts of wildfire does EJ literature highlight? Or perhaps what distinguishes EJ literature on human impacts of wildfire from the other non-EJ literature? The authors could give some categories and some examples.

Line 430: “Our findings identified disparities in the foci of publications in terms of effects, community types, and time periods,”

I did not see any analyses or reporting of “effects” found in the selected studies.

Discussion

“Conversely, human dimensions literature focused on events occurring during (e.g., evacuation) and after wildfires (e.g., recovery and adaptation) were comparatively less studied.”

It is difficult for me to believe this. There may be hundreds if not thousands of studies on the social, economic and health impacts of wildfire. I cannot tell why the authors came to this conclusion because the selected literature are not listed anywhere in the paper, and I do not know which databases the authors searched.

For this reason, further statements or conclusions about “amount of research” on various topics, community types, and EJ-focused studies, are suspect.

If the authors included only English-language publications, then it is not worth assuming that there has been no research on social dimensions of wildfires outside of English-speaking countries.

It is difficult to tell what the purpose of the EJ discussion is. For example:

“Also, overall EJ relevant studies significantly focused on race/ethnicity and poverty aspects and to a lesser degree on employment, gender, and language (Supporting Information 4).”

Race/ethnicity and poverty aspects of what? Disaster planning? Social and economic impacts of wildfire?

6. PLOS authors have the option to publish the peer review history of their article (what does this mean?). If published, this will include your full peer review and any attached files.

Reviewer #1: No

Reviewer #2: No

Reviewer #3: No

Reviewer #4: No

---

## [Author Response · Author response to Decision Letter 0]

31 Mar 2022

PONE-D-22-00418

A burning issue: Systematically reviewing the human dimensions and environmental justice aspects of the wildfire literature PLOS ONE

Dear Dr. Escobedo,

Thank you for submitting your manuscript to PLOS ONE. After careful consideration, we feel that it has merit but does not fully meet PLOS ONE’s publication criteria as it currently stands. Therefore, we invite you to submit a revised version of the manuscript that addresses the points raised during the review process.

Please include the following items when submitting your revised manuscript: A marked-up copy of your manuscript that highlights changes made to the original version. You should upload this as a separate file labeled 'Revised Manuscript with Track Changes'.

Julia A. Jones

Academic Editor

PLOS ONE

Additional Editor Comments:

All four reviewers saw potential in this paper, but also expressed concerns. Please address the following issues in your revision: 

(1) explain why this topic is important and to what audiences, 

Response: We hope that our revisions in the Introduction, Discussion, and Conclusion sections of our manuscript, as well as responses to all 4 reviewers, are now communicating the importance of this topic. Regarding our intended audience, as you will see, it was targeted towards those working in the areas of geography, social sciences, environmental sciences and ecology, and biodiversity & conservation; rather than those in public and environmental and occupational health. We note that this revision also has direct implications for your 2nd comment/suggestion that follows. Please see below in the response to the 4 reviewers for specific examples of our revisions. 

(2) consider including the health impacts of proximity to both wildland fire and wildland smoke, which is a key element of justice, 

Response: As shown in the previous response regarding our intended audience and results from analysis we did not include in the original manuscript (Supplementary Information Fig 1); a review of the literature around the health effects of wildfires and wildfire smoke was outside the aim and scope of our manuscript and was a distinct subject area separate from the social-demographic space our study focuses on. As you will see in the below responses to the reviewers, there are also already multiple reviews (systematic and scoping) of the literature around the health effects of wildfires. Similarly, there is considerable literature on the environmental justice aspects of air pollution. We have however included several more sentences on this subject in the discussion to justify our reasons for not doing this. Please see the specific response below for more examples. 

(3) summarize major findings from this literature and what has been learned, 

Response: Thank you and the reviewers for this comment since it has given us the opportunity to revise our manuscript. To address these concerns, you will see below, that we have added several new paragraphs in the Discussion section that highlight some of the emerging fields and literature that considers EJ. In this new text we took considerable effort to highlight some of the state of the art and seminal literature in this field of research. Please see our responses to specific reviewers for specific examples of our revisions. 

(4) accurately select and define the type of review that this paper contains.

Response: We believe that our manuscript has elements of both a scoping and systematic literature review. Indeed, it is similar to other systematic reviews in the natural disaster, environmental science, ecology and biodiversity conservation fields. We also read the PLOS One guide to authors which states a systematic review is “…a review of a clearly formulated question that uses explicit, systematic methods to identify, select, and critically appraise relevant research, and to collect and analyze data from the studies that are included in the review.” The guide then goes on to say, “Statistical methods (meta-analysis) may or may not be used to analyze and summarize the results of the included studies.” Accordingly, we developed three specific research questions that were addressed using systematic methods. As an example, we used a probit model to statistically analyze articles we systematically coded for EJ relevance and to identify social-demographic variables. However, given the concerns raised by reviewer 4 around the use of ‘systematic review’, we have changed our review type to “literature review” throughout the manuscript. Please see below for specific responses to the relevant reviewer(s). 

Comments to the Author

1. Is the manuscript technically sound, and do the data support the conclusions?

Reviewer #1: Partly

Reviewer #2: Yes

Reviewer #3: Yes

Reviewer #4: No

2. Has the statistical analysis been performed appropriately and rigorously? 

Reviewer #1: I Don't Know

Reviewer #2: Yes

Reviewer #3: I Don't Know

Reviewer #4: I Don't Know

3. Have the authors made all data underlying the findings in their manuscript fully available?

Reviewer #1: Yes

Reviewer #2: Yes

Reviewer #3: Yes

Reviewer #4: No

4. Is the manuscript presented in an intelligible fashion and written in standard English?

Reviewer #1: Yes

Reviewer #2: Yes

Reviewer #3: Yes

Reviewer #4: No

5. Review Comments to the Author

Reviewer #1: This paper is really nicely done from a professional standpoint - rigorous analysis, nicely written, etc. I had only some minor suggestions and then a couple of larger concerns:

Lines 180-3 – probably would be wise to not use the word “injustice” in the definition of justice.

Response: Done. We have replaced “... not unjustly/disproportionally...” with “…inequitably…”. 

Lines 498-500 – this seems almost tautological.

Response: Yes, we see your point. So, we have revised this text and it now reads, “Indeed, these costly and highly severe wildfire events are reactively resulting in greater and subsequent scientific productivity in certain regions and countries on these topics particularly related to response; as opposed to proactively investing and studying planning and response related issues (United Nations Environment Program, 2022)” (See 6th paragraph of the Discussion section). We hope this clarifies the text.

Line 513-15 – I’m not sure the findings support this claim, simply identifying a gap or an imbalance doesn’t prove the need to fill that gap or remedy that imbalance. There is at least one missing premise in the argument here – maybe about some harm that this gap or imbalance creates?

Response: Thank you for catching this. We have revised this text and added a United Nations report to substantiate our revised text which now reads, “These findings confirm the need for continued research not only in higher income countries frequently affected by wildfires, but also other regions in medium and low-income countries with increasing occurrence of wildfires that may not receive as much international attention (United Nations Environment Program, 2022)” (See Lines 578-583).

Line 517 – this is an unnecessary speculation.

Response: We agree with this comment and have removed the “perhaps indicative of ‘parachute science’” text in the revised manuscript.

Lines 517-19 – while I agree that there are benefits to such collaborations, the argument for them needs to be made independently – I’m not sure the speculation about lack of co-authors is enough to support this argument.

Response: Our graphical and bibliometric findings regarding co-authorship and lack of geographic representation of study areas, we feel, make this statement more a finding than a speculation. However, we see your point. So, to address your comment, we have moved this text to the first paragraph in the Conclusion (Lines 765-771) to make this more of a recommendation than the discussion of a finding. The text now reads, “First, we note that our review methods omitted some relevant global literature since it was limited to English language publications indexed in available databases. This means that many relevant publications from frequently wildfire affected Mediterranean and Latin American regions were likely excluded. This suggests that in seeking to expand the geographic scope of the social-demographic aspects of wildfire literature, researchers from higher income countries might benefit from greater collaboration with scientists from the Global South…” 

I appreciate the suggestions for future research.

Response: Thank you. 

The analysis would benefit by reference to some parallels. I’m not sure what to make of X% of some body of literature mentioning EJ. Is that high, low, average? Could this be compared to some other body of literature? Maybe others have done something similar in other fields, e.g., wildlife.

Response: Thank you for this suggestion. We decided that comparing it to the body of literature on wildfire smoke and health effects would make for a useful comparison, accordingly on Lines 599-606 we have included new text that states, “However, in their review of the physical health effects of non-occupational wildfire smoke, Liu et al. (2015) found that only three out of 61 (5%) studies reported “effect modification by socio-economic status, race, or co-morbidities” (p.128). Similarly, Reid et al. (2016) reviewed 115 articles on the health impacts of wildfire smoke exposure and noted that “Few epidemiological studies have investigated whether specific populations are more susceptible to wildfire smoke exposure than the general population” (p.1340). This suggests that the EJ aspect of wildfires could be more prominent in the socio-demographic dimensions versus the health effects literature.”

I think the paper would benefit from an attempt to explain to the reader why this matters. Why does it matter that EJ is discussed in some percentage of articles in a given literature, or maybe why does it matter that EJ is discussed at all? An answer to this question, even a modest one, would make the paper much more engaging.

Response: Thank you for the suggestion and the opportunity to improve the reach of the paper. We have added the below paragraph to explain at least three audiences that would benefit from our work. The three audiences we identify here include land managers, policy makers, and researchers in the fields of: geography, social sciences, environmental sciences-ecology, and biodiversity-conservation. 

“By providing insights into the timing and scope of the literature on this issue, we are able to help guide future work in this field. This is particularly important to land and environmental managers who are working to incorporate social justice (i.e., EJ) considerations into management practices as well as policy formulation and uptake. Further, this work expands the conversation on wildfire to include the social-demographic and economic impacts as well. Explaining and quantifying the ways in which community differences are included in the literature allows researchers to better understand how these natural disasters distribute damages disproportionally, not only across landscape, but also across communities and society. In doing so, practitioners working in conjunction with policy makers can craft more socially equitable policies (e.g., home insurance, community recovery programs) as presented in this study.” (lines 132-142)

Furthermore, to better address the importance of EJ in literature, we have added the following text, “Understanding EJ as applied in our definition, can help guide future research and policies in this area. As climate change impacts increase throughout the world, more communities will be affected by extreme events such as wildfires. However, these effects are unequal as the most vulnerable communities (i.e., EJ) usually suffer from the strongest impacts; and many of these communities do not have the resources to mitigate, recover, or adapt from these effects. Therefore, policy makers need to identify barriers vulnerable communities face to better understand how to develop strategies to help reduce impacts from climate change and increased wildfire occurrence and severity.” (Lines 592-599)

Reviewer #2: This manuscript presents a systematic review of the EJ literature related to wildfire. The new, wildfire-specific definition of EJ presented was effective and should be shared for more broad use across the interdisciplinary field.

Response: Indeed, one of our intended purposes is to present readers with an evidence-based definition that can be readily used by not only applied, interdisciplinary researchers, but by land management and environmental organizations in their planning and policy documents. 

This manuscript addresses three stated objectives to 1) determine what aspects of the human dimensions of wildfire are most frequently studied, 2) identify the geographies and contexts that have been the focus of these studies, and 3) given 1-2, what aspects of environmental justice are being addressed in the human dimensions of wildfire literature. ….While I agree that these objectives were all addressed, I was surprised at the lack of information that was presented regarding the health impacts of proximity to both wildland fire and wildland fire smoke. This gap was made clear in the fact that none of the top ten cited journals were in health fields. As a growing public health concern, I was surprised by the lack of public health data presented. I think this manuscript is complete as is. However, I think it would add to the discussion if this were addressed as an indication that wildland fire research and public health research need to be brought together to address this growing interdisciplinary crisis.

Response: We appreciate these comments and suggestions and we do see you point. But, in the following response we will provide four justifications for not including such wildfire smoke and health literature in this manuscript. 

First, as we were analyzing our literature review data, we developed a dendrogram which was not included in our original manuscript, but is now included as Supporting Information Figure 1. In it we saw that the wildfire smoke/air quality literature was a distinct and separate topic (i.e., the distinct and separate blue colored branch in SI Fig 1) and as such was outside the scope of our manuscript’s intended audience (See the following response for the specific fields) and these terms were therefore not used.

Second, we chose to target our paper towards audiences in the fields of: geography, social sciences, environmental sciences, ecology, and biodiversity-conservation; rather than those in public health and the environmental and occupational health areas. This and # 1, meant that we did not include search strings such as “wildfire and health” or “Wildfire smoke and air quality” as part of our methods. However, to clarify this our revised manuscript now makes this clear by stating, “Wildfire AND health” or “Wildfire AND smoke AND health”; and other similar terms, were not used as search terms due to the availability of existing reviews on both the health effects of wildfires (Black et al., 2017; Liu et al., 2015; To et al. 2021) and the EJ aspects of air pollution (Fairburn et al., 2019; Ferguson et al., 2020) as well as findings from our analysis indicating they were a different and distinct topic (Supporting Information Fig 1).” (lines 239-244).

Thirdly, there is already a substantial body of research around air pollution and environmental justice, as well as wildfires and human health, including reviews of both subjects. This meant that there was little need for us in repeating what has already been reported in this work. The following is a non-exhaustive list of such reviews on these topics:

Reviews on air pollution and EJ

 Fairburn, J., Schüle, S.A., Dreger, S., Karla Hilz, L. and Bolte, G., 2019. Social inequalities in exposure to ambient air pollution: a systematic review in the WHO European region. International journal of environmental research and public health, 16(17), p.3127.

 Ferguson, L., Taylor, J., Davies, M., Shrubsole, C., Symonds, P. and Dimitroulopoulou, S., 2020. Exposure to indoor air pollution across socio-economic groups in high-income countries: A scoping review of the literature and a modelling methodology. Environment international, 143, p.105748.

 Hajat, A., Hsia, C. and O’Neill, M.S., 2015. Socioeconomic disparities and air pollution exposure: a global review. Current environmental health reports, 2(4), pp.440-450.

 Miao, Q., Chen, D., Buzzelli, M. and Aronson, K.J., 2015. Environmental equity research: review with focus on outdoor air pollution research methods and analytic tools. Archives of environmental & occupational health, 70(1), pp.47-55.

Reviews specific to wildfire smoke and health

 Black, C., Tesfaigzi, Y., Bassein, J.A. and Miller, L.A., 2017. Wildfire smoke exposure and human health: Significant gaps in research for a growing public health issue. Environ Toxicol Pharmacol, 55, pp.186-195. https://doi.org/10.1016/j.etap.2017.08.022

 Grant, E. and Runkle, J.D., 2021. Long-Term Health Effects of Wildfire Exposure: A Scoping Review. J Clim Chang Health, p.100110. https://doi.org/10.1016/j.joclim.2021.100110

 Groot, E., Caturay, A., Khan, Y. and Copes, R., 2019. A systematic review of the health impacts of occupational exposure to wildland fires. Int J Occup Med Environ Health, 32(2), pp.121-140. https://doi.org/10.13075/ijomeh.1896.01326

 Isaac, F., Toukhsati, S.R., Di Benedetto, M. and Kennedy, G.A., 2021. A Systematic Review of the Impact of Wildfires on Sleep Disturbances. Int J Environ Res Public Health, 18(19), p.10152. https://doi.org/10.3390/ijerph181910152

 Liu, J.C., Pereira, G., Uhl, S.A., Bravo, M.A. and Bell, M.L., 2015. A systematic review of the physical health impacts from non-occupational exposure to wildfire smoke. Environ Res, 136, pp.120-132. https://doi.org/10.1016/j.envres.2014.10.015

 Reid, C.E., Brauer, M., Johnston, F.H., Jerrett, M., Balmes, J.R. and Elliott, C.T., 2016. Critical review of health impacts of wildfire smoke exposure. Environ Health Perspect, 124(9), pp.1334-1343. https://doi.org/10.1289/ehp.1409277

 To, P., Eboreime, E. and Agyapong, V.I., 2021. The impact of wildfires on mental health: a scoping review. Behav Sci, 11(9), p.126. https://doi.org/10.3390/bs11090126

However, based on your suggestion and that of the Editor, we have added: 1) a dendrogram (SI Fig 1) showing that the wildfire smoke/air quality literature was a uniquely distinct and separate topic from our study’s aim and intended audience and 2) we added some additional text to the discussion around this subject and our findings. It reads, “The results of our review therefore suggest that a more interdisciplinary research approach to wildfires might be needed to address this growing crisis. The effects to communities from wildfire, and their preparation for such events, should include not only wildfire and subsequent smoke effects, but other potential secondary effects such as flash floods.” (lines 534-538)

In reference to your comment about the journals, you are correct. The top four journals for the health effects of wildfire smoke research appear to be: Environmental Health, Environmental Health Perspectives, Environmental Toxicology and Pharmacology, and International Journal of Environmental Research and Public Health. But, as previously stated, these journals are all intended for audiences in the public health as well as the environmental and occupational health fields; which again are not our intended audience. We hope the above addresses your concerns. 

Reviewer #3: PLOS 1 Review: 1 February 2022

This topic is timely, and the title is intriguing. My first impression is that the MS would discuss substantive findings from the relevant literature. I was hoping to learn more about how EJ factors affect responses to wildfire/prescribed burns. 

Response. Thank you, please read the following responses and revisions. Hopefully, we have now addressed these concerns in this revised version. 

Rather, the MS analyzes components of that framework in detail. This is okay from a technical or descriptive standpoint, but the piece would be more inviting if it summarized major findings from this literature—if it told the reader how environmental justice indicators like race or income or language made for unjust situations. What have we learned so far from this nascent literature? This is what’s really important. This could be summarized by country, region of country, or some other way. I think you can keep all you’ve done but add to it summaries of findings from this literature.

Response: Thank you for the opportunity to add and improve our manuscript. To address your concerns, we have written an additional 3 paragraphs of revised text in the discussion that summarize some of the emerging fields that consider EJ. This highlights some of the state-of-the-art subjects and results currently in this field of research. Here, we provide examples of some of this new and revised text and citations found in lines 682- 728:

 “The literature reviewed and findings in this study served to summarize the most relevant articles in several important areas related to EJ and wildfire. These include studies on fuel treatments, exposure to fire risk, adaptive capacity, defensible space, and environmental justice broadly. 

 ….Winter et al. (2002) found that agency trust is an important factor in the acceptance of fuel treatments. Similarly, Mylek and Shirmer (2019) …. found that higher income households are more accepting of prescribed burning, thus reinforcing the result from Winter et al. (2002) that agency trust is important. 

 We also found that most fuel treatment literature understandably focused on WUI areas, and that the studies that focused on WUI areas had a lower likelihood of being labeled as EJ-relevant as seen in Table 4. Although Figure 6 shows that prescribed fires are still considered a basic theme, there is still work needed in this field. 

 …another body of reviewed literature concerns itself with homes (e.g., defensible space activities) and community investments for wildfire mitigation. Carrol and Paveglio (2016) highlight that community characteristics are important for investments for fire mitigation behavior, …. Mockrin et al. (2015) found that home rebuilding was a slow process and that home and parcel-level investments for post-fire mitigation were low. 

 Thus as corroborated by our literature review, social-demographic characteristics are important for communicating the risks effectively (Meldrum et al. 2021). 

 Another body of literature as shown in Figure 6 concerns environmental justice. 

 studies from our review that focused on EJ describe differential impacts on different communities. For example, Masri et al. (2021) found that with the exception of Native American communities, wildfires impact census tracts with lower populations of minorities and higher populations of elderly. However Native American communities are much more likely to be impacted by wildfire effects.”

Please see this section for the revised text in these 3 new paragraphs.

Page 6, lines 106-108. This isn’t always the case. Poor people do live in fire prone areas. I don’t know whether you reviewed Timothy Collins’ papers from the mid-2000s which discuss poor people made more vulnerable to wildfire when rich people moving to ecologically fragile places in the WUI:

 Collins, T.W., 2005. Households, forests, and fire hazard vulnerability in the American west: a case study of a California community. Environmental Hazards 6, 23–37.

 Collins, T.W., 2008a. The political ecology of hazard vulnerability: marginalization, facilitation and the production of differential risk to urban wildfires in Arizona's White Mountains. Journal of Political Ecology 15, 21–43.

 Collins, T., 2008. What influences hazard mitigation? Household decision making about wildfire risks in Arizona's White Mountains. The Professional Geographer 60, 508–526.

 There are also two papers from the southeastern US that analyze the intersection of social vulnerability and wildfire risk: Poudyal et al, 2012 (Environmental Management) and Gaither et al, 2011 (Forest Policy & Economics). 

 Gaither et al 2020 also considered exposure to prescribed burn smoke as an injustice—as opposed to Adams & Charnley who looked at the absence of prescribed burning as the injustice.

Your governance/information is similar to existing EJ definitions that describe it as including ‘procedural or participatory’ elements. 

 See Setha Low 2012 for a full discussion. Public space and diversity: Distributive, procedural and interactional justice for parks. In G. Young, & D. Stevenson (Eds.), 

 The Ashgate research companion to planning and culture (pp. 295–310). I think you should acknowledge this and literature that informed the components you include in your definition.

Thank you for your comment and suggestions for papers to review. However, Gaither, Adams & Charnley, Collins 2005 and Poudyal et al, 2012 were all identified and have been accounted for in our literature review, as was another article by Collins (2009); which is similar to Collins 2008a and 2008b. So, we agree that poor people do live in fire prone areas, and as has been reported, greater numbers are now increasingly living in these areas. In fact, the Collins (2008) article you mention states “Most previous studies have not closely considered the coexistence of wildfire hazards, poverty, and social vulnerability …”. Unfortunately, with this comment you are referring to, we were strictly referring to the Wigtil et al. (2016) and Davies et al. (2018) articles, and as such we obviously did not communicate this effectively. 

So, to address this comment we have now revised this text to read, “Conversely, these authors report that lower income and minority populations have been largely reported to be concentrated in areas historically at lower risk, such as in city cores and highly urbanized suburbs.” (lines 110-112). We also noted how we have begun to see shifts in this, where lower income people are driven into fire-prone areas for reasons such as affordable housing. Accordingly, we have also added another revised sentence in response to your suggested papers, “Amenity migration by higher income populations to peri-urban areas also puts existing residents, that are often vulnerable populations, at risk in those communities through a larger number of homes to defend in case of wildfire, since many newer residents choose not to engage in fuel reduction projects (Collins, 2008).” (lines 116-120). This is in addition to the existing sentence that states, “However, recent housing price increases in urban areas of North America (Greenberg, 2021), increases in peri-urban, less formalized settlements in Latin America, Africa, and Asia (Williams et al., 2019) and the arrival of immigrants in the Mediterranean region are changing the demographic and socioeconomic make-up of the WUI and communities at risk from wildfire (Shaw et al., 2020).” (lines 112-116).

In regard to the sources that informed our definition and components of EJ, we already had noted the “governance” related components of EJ according to Adams and Charnley, “Adam and Charnley (2020), who studied the EJ of hazardous fuel treatments in US National Forests, categorized the concept into three themes: (1) social vulnerability and resilience in relation to wildfire risk management; (2) EJ and federal agency natural resource management; and (3) participation of EJ populations in collaborative resource management by US federal agencies.” Similarly, the EPA’s definition of environmental justice also contains a participatory element “and equal access to the decision-making process to have a healthy environment in which to live, learn, and work” (EPA, 2021).” (lines 177-181). Again, we note that our “governance” component in our methods accounted for all these government-society and planning/participatory/decision-making/policy processes.

However, to make it clearer that we have drawn upon this literature in defining EJ, we add revised text and referenced highly cited publications on EJ. The revised text and citations now read, “Based on the above literature (Adam and Charnley, 2020; EPA, 2021) as well as other EJ definitions (e.g., Agyeman et al., 2016; Schlosberg, 2007) …” (lines 194-195). As for the Setha Low 2012 reference suggested, it is focused on urban public parks and is thus not relevant for our definition of EJ.

Your EJ definition on p 9 comes off as somewhat top down and authoritative. It dictates where people can live without acknowledging that people are sometimes willing to assume risk. The definition should be reworded in positive terms, that is, in a way that says what EJ is and less what it is not. It should not suggest that people don’t have a choice in where to live.

This is an interesting point and interpretation of our definition you bring up. To specify, our definition, as applied and assessed in this study and as previously mentioned in response to Reviewer 2’s initial comment, is meant to be used not just by researcher, but for environmental and conservation policy formulation and uptake by regulatory environmental/land management organizations. We understand that because of this emphasis, it can come across as top-down. However, this phrasing is typical of most policy instruments that we are aware of, and as a result we find it to be a more useful approach compared with other alternatives for phrasing it. But, hopefully we responded, in part to this and in part to another reviewer’s comment, by removing the word “unjustly” from the definition. 

However, we respectfully disagree with you regarding the use of “choices” and “positive terms”. The definition of Environmental Justice as used in this paper is a framework to understand the inherent lack of choices, and limitations, encountered by historically disadvantaged communities. As stated in the definition you are referring to, the communities under consideration “they ...have not been historically engaged, consulted, and meaningfully involved in governance processes that affect their environment…”; and as a result, their choices of where to live, engage in land management, etc. may be affected. This can occur both through the limiting of their choice set, as well as making some choices more costly than others. In the context of fire risk, there are several factors that either incentivize or have historically forced individuals (i.e., Native American) within EJ communities to select more fire prone communities. 

For example, as stated previously in the manuscript (lines 90-91), many people cannot afford homes in some communities due to real estate premiums in urban core areas and are thus increasingly moving into areas of higher fire risk. Also, as Davies (2018) found, Native Americans are overrepresented in areas with high wildfire hazard, yet they largely did not choose to live in these locations having been forced onto reservations. Furthermore, EJ communities in fire-prone areas may not understand the real fire risk, how to reduce or mitigate this risk; or have the capacity to do so (i.e., many suggested measures to fireproof a home require financial investments, sometimes major ones). So, this is key wording in our definition and the whole premise of EJ. Given such disparities and disenfranchisement, we find it difficult to strike a positive chord when it comes to such unfair and inequitable realities. 

Page 30, lines 642-643: not sure what this means: “Thus, future research on the socio-ecological dimensions of wildfire would benefit from focusing on human population-based variables and their dynamics as well….”

Response: Thank you for highlighting this. We have revised this text and it now reads, “Thus, future research on the socio-ecological dimensions of wildfire would benefit from including specific socioeconomic and demographic variables and metrics identified in this study in their analyses as well as a broader geographic scope outside the US.” (lines 776-778)

Reviewer #4: Review comments on PONE-D-22-00418

This manuscripts describes a review of literature on what the authors call the human dimensions of wildfire. It is intended to be a survey of the content and geographic focus of the literature, as well as the extent to which environmental justice as a framework has been applied to this research. Unfortunately, I found the scope and research questions to be very unclear, the methods very unclear, and the results and discussion to lack clear (or useful) framework or message.

Response: It is unfortunate that these were the Reviewer’s conclusions, but we note that they differ greatly from the other 3 reviewers. That said, we respect your comments, and we will address your concerns below point by point. And you will see, we have made revisions where necessary to address as many of your concerns as possible. 

A main critique is that this study should not be described as a systematic review. The study aims and questions are very general (and somewhat vague), and the goal is not to compile and compare studies using a specific study design, experimental treatment or technique, Cochrane defines systematic reviews as: “A systematic review attempts to identify, appraise and synthesize all the empirical evidence that meets pre-specified eligibility criteria to answer a specific research question. Researchers conducting systematic reviews use explicit, systematic methods that are selected with a view aimed at minimizing bias, to produce more reliable findings to inform decision making.” I suggest that the authors review the wide selection of writing about different review types and select a more appropriate review definition. Scoping review might be a good selection.

Response: A research study’s aim should rarely be to simply implement methods; it should be to guide the use of methods to address/test specific research questions. That said, although much of the study is more exploratory in nature, our original manuscript articulates an aim and 3 specific research questions that are presented in Lines 216-222. Therefore, we feel we cannot be clearer in stating, “Therefore, the aim of this study is to use environmental justice as a lens to better understand the state of the art of the role of wildfire in communities and human settlements. Specifically, this study had three objectives: 1) determine what aspects of the social-demographic dimensions of wildfire are most frequently studied; 2) identify the geographies and contexts that have been the focus of these studies; and 3) given 1-2, what aspects of environmental justice are being addressed in the social-demographic dimensions of wildfire literature.”

Furthermore, we are aware and have read multiple published papers, as well as co-authored manuscripts, that are called systematic reviews and have aims, methods, and a format similar to ours published in PLOS One and elsewhere. Most importantly we read and applied PLOS One’s guidelines -not Cochrane’s - to authors which states a systematic review is “..a review of a clearly formulated question that uses explicit, systematic methods to identify, select, and critically appraise relevant research, and to collect and analyze data from the studies that are included in the review.” We have done this with our coding of the literature and the use of both bibliometric analyses and a probit model and chi-square test to statistically analyze our identified literature. The guidelines also say: “….statistical methods (meta-analysis) may or may not be used to analyze and summarize the results of the included studies”, suggesting that we do not need to use statistical methods for every aspect examined. Below in our other responses, we also detail in full transparency and repeatability, how we carried out and analyzed our literature review. 

However, to address your critique, we have decided to strike “systematic literature review” and now refer to the study in the revised manuscript as a “literature review’. We have changed this throughout the manuscript. 

Other specific concerns are listed below.

The authors use a lot of jargon coming from forestry (I believe). For example, while the authors spell out the acronym WUI, they never define it. What is the WUI and why is it an area of relevance in relation to wildfire ecological and social impacts? Same for “peri-urban”

Response: We are not sure what the Reviewer means by “..jargon coming from forestry”? Here, and elsewhere, the Reviewer seems to suggest that we delve into, and define, each and every concept we use within the first 2 pages of our manuscript. We find this difficult since we first have to provide the necessary background and justification for the study. And only then can we being to define concepts and how they relate to our study aim and objectives. As previously explained to the Editor and another reviewer, our intended audience is from the geography, social sciences, environmental sciences and ecology, and biodiversity & conservation areas. So accordingly, we must use terms and concepts from these field as opposed to say “jargon” from the health or engineering fields. However, to address your concern regarding WUI, and to not distract readers from our stated objectives, we added the following text in Lines 85-86, “See Stewart et al., (2007) for a discussion of the definition and application of WUI in wildfire management and policies”. We note that the term WUI was, and is, specifically defined later in the Discussion section (Lines 658-661 of the revised manuscript).

On line 85 the authors switch the focus to WUI, while it is not clear why.

Response: We feel that the previous paragraph lays out that WUI and urban areas are where most people live (56% of the world’s population) and hence is the key socio-demographic and ecological context affected by wildfire- thus its importance. As such the paragraph beginning in line 85 that you refer to is where we then begin to document the importance of the WUI regarding wildfire effects and impacts to communities.

I’m curious, if the authors wanted to include literature on the human health impacts of wildfires and smoke from wildfires, why they did not include search term “health”?

Response: Thank you for this comment. We have addressed this in our response to the Editor and Reviewer 2. But basically, there are 4 reasons why we did not include the human health impacts of wildfire. First, a dendrogram (now as Supporting Information Figure 1) of our study revealed that wildfire smoke/air quality research was a distinct topic that is separate from all the other study topics. Second, we chose to target our manuscript towards readers working in the fields of geography, social sciences, environmental sciences & ecology, and biodiversity & conservation; not those from the public health and environmental-occupational health fields. Hence, because of #1 and #2, we did not include search strings such as “wildfire and health” as part of our methods. Third, there is already a substantial body of research around air pollution and environmental justice, as well as wildfires and human health, including reviews of both subjects. This meant that there was little justification for us to repeat this work. Please see our response to the Editor and Reviewer 2 for specific responses and revisions we made to clarify this. 

Finally, we also note that the quantity of literature on this additional and non-relevant subject would make reading and coding all the relevant literature prohibitive. Specifically, a Web of Science search for “wildfire and health” resulted in 1810 results, and 1610 in Scopus. On PubMed, which would be the most appropriate database for this topic, the same search string gives 835 results. Given that we had to read and code every identified article, and that this project’s constraints, we would have not been able to carry out the research. And again, the purpose of doing this would have been beyond the scope of our literature review as explained above. 

The authors use the term “environmental justice first on line 113, but without defining the term “As such, the increasing vulnerability of peri-urban and WUI areas, and newer socio-demographic groups, to wildfires indicates a need for environmental justice research in this area.”……. The 2nd use of the word is in the statement of study aims: “Therefore, the aim of this study is to use environmental justice as a lens to better understand the state of the art of the role of wildfire in communities and human settlements.” This is a big surprise because the authors have not yet defined the term or how it could be useful as a lens.

Response: This is only our 4th introductory paragraph and third page, so it is difficult to define this and all other concepts at this point in our manuscript? However, please see the following response to see how we addressed your concern. 

1.3 Aims and objectives

In this section, I do not find any aims or objectives stated.

Response: Apologies, we moved our stated aim and objectives from the fifth paragraph in our Introduction, to this section in Lines 217-223 and it is now presented only after EJ is defined (lines 154-192).

Methods

The selection criteria is extremely vague: “Selected reports that were not filtered out had to meet two criteria: 1) explicit focus on wildfires (including prescribed burns, risk, and defensible space); and 2) discuss the human dimensions aspect (e.g., adaptive capacity, sociodemographic variables).”

Response: We apologize if these were vague to you; however, the aim and scope of our study means that eligibility criteria were, by definition, quite broad. However, we have revised the criteria to make this clearer, “Selected reports had to meet two criteria: 1) explicit focus on wildfires (including prescribed fire, risk of wildfire, defensible space/wildfire mitigation, wildfire smoke, secondary effects of wildfires such as mudslides); and 2) discuss people and/or human communities (e.g., ability to prepare for and recover from wildfires, impacts of wildfires on people/human communities, sociodemographic variables such as income and race).” (lines 252-257)

What is “the human dimension aspect”? 

Response: We have replaced “human dimensions” with “socio-demographic” in the title and elsewhere throughout the manuscript. 

What is adaptive capacity? Variables are elements of statistical analyses, not topics of study.

This is a critical flaw of the study. There is no way that any other person/group could reproduce this study, which is the main criteria of a systematic review (and any review).

Response: We have defined adaptive capacity as “(..the ability to prepare for and recover from wildfires” (lines 255 - 256) and we have replaced “variables” with “characteristics” in Line 257.

However, we disagree with the reviewer that this is a critical flaw in our study. Indeed, there are multiple examples of literature reviews and meta-analyses where “variables” are information and metrics gleaned from the identified literature and that are then used for “elements of [a] statistical analyses” as is the case of our probit model. We also strongly disagree that our manuscript was neither transparent nor repeatable. We will not repeat our methods here but throughout Section 2 of our submitted manuscript we report all aspects of our methodology based on PLOS One guidelines. But, for example here are some methods:

 We used the Preferred Reporting Items for Systematic Reviews and Meta-Analyses (PRISMA) standards (Page et al., 2021) to guide our reporting of the …. review. 

 We first used the Scopus database to search for peer-reviewed scientific articles and reviews written in English matching pre-determined key words and phrases.

 See Table 1.

 …we combined the results and used a Quality Assurance and Control (QA/QC) process to remove duplicates (356) as well as book chapters and publications not in English (n=16). Next, titles were screened for relevance to the research objectives.

 Another 359 reports were eliminated after a full reading; leaving 182 studies to be included in the ….review.

 we also searched the Web of Science Core Collection database using the same search terms(Table 1) and protocols.

 All selected publications in our final database were then individually read and coded following a protocol of pre-determined attributes based on our overall literature review and study objectives (Supporting information).

We can go on, but we urge you to please reread the methods section. And, if you have more specific recommendation on how to make our methods more repeatable, please let us know. 

The selection criteria should be explicitly and clearly defined in the main manuscript, not put in an appendix. If the authors need to make space, the selection criteria is far more important than list of journals presented in Table 2.

Response: We agree with this comment and have now put this in the main manuscript in the text and as Table 2, “The time period(s) of the study was classified as at least one of four options: before wildfire/prescribed fire (i.e., plans to evacuate, preparing the home for wildfire), during wildfire/prescribed fire (i.e., evacuation during a fire), post-wildfire (i.e., recovery after a wildfire), and modeled wildfire. We also coded the studies for what aspect(s) of wildfire (e.g., wildfire, smoke, floods) was explored under hazard, with prescribed burning and mechanical thinning coded under “other”. For the EJ attribute, we classified the components as explained in section 1.2 to provide a more detailed statistical analysis of our study objectives. Finally, the socio-demographic category was coded to specify different sociodemographic variables used in the relevant literature (e.g., age, gender, language). Importantly, only including the attribute(s) when discussing the study community/sample population was not sufficient to be coded “yes”. Instead, the influence of the attribute needed to be examined to code the study “yes” for the sociodemographic variable.” (lines 283-295)

And, in lines 201-216 we now state “Additionally, to better and quantitatively analyze the disparate socio-demographic dimensions literature, we classified EJ as being characterized by four components: 

 EJ-Impacts/Harm: No disproportionate impact from environmental harm on disadvantaged communities or individuals. Examples: Location. i.e., disadvantaged communities forced to live in a high-risk fire zone, in an area/home that will suffer from prolonged exposure to fire-related smoke/floods, or live-in fire prone areas that have not had recent prescribed burns or fuel treatments

 EJ-Governance: equal and meaningful access to environmental information and participation in decision making. Examples: transparent access to culturally relevant information on fire-proofing your home, participation in land management and planning 

 EJ-Amenities: Equal access to environmental amenities such as clean drinking water, sanitation, and parks. Examples: Access to good quality air, clean water after a fire

 EJ-Remedies: Access to justice and effective remedies for environmental harm. Examples: Post-fire assistance/aid, access to new/temporary housing post-fire” 

The authors state nowhere which databases they searched. Therefore I cannot tell what biases might have been introduced in the search, and there is no way that someone could reproduce the author’s findings.

We are perplexed with this statement. In Lines 197 (page 10) of our original submission, we specifically and clearly state “…We first used the Scopus database to search for peer-reviewed scientific articles and reviews…”. Then in line 221 (page 11), “we also searched the Web of Science Core Collection database using the same search terms...”

Regarding repeatability of methods, we urge the reviewer to read Section 2.1 where we systematically describe our methods in detail per PLOS One guidelines to authors. Figure 1 also details the number of reports eliminated at each stage in accordance with PRISMA guidelines (as required by PLOS One). We are unsure how we can be any more specific and transparent than this regarding databases and methods.

Results

Section 3.1- Which studies and how many were included in the analysis?

Response: Again, this was clearly stated in lines 280-281 (page 14) of the original manuscript “In total, 299 studies were selected for analysis: 275 original research articles, 17 reviews, 5 reports, 1 letter, and 1 note.” Figure 1 also clearly shows 299 studies included in our analysis.

On line 217 is the statement “Another 359 reports were eliminated after a full reading; leaving 182 studies to be included in the systematic review.” But in the Results it states “In total, 299 studies were selected for analysis”. In the References list, there are only about 80 references listed. This is very unclear.

Response: We apologize if this was unclear. The 182 studies referred solely to those selected from Scopus, the first of two databases searched. We have now clarified that, “… leaving 182 studies from Scopus to be included in the systematic review.” (lines 263-264). We also refer the reviewer to Figure 1 which shows the number of studies excluded at each stage of the screening process, and the final number of reports included. Due to the large number (n=299) of studies included in the review, we were unable to list them all in a table within the manuscript. Instead, our data will be publicly available on ICPSR (https://www.icpsr.umich.edu/web/pages/) after publication of this manuscript. This will allow readers to view the full bibliographic information for all 299 studies.

It’s not clear why the journals in which selected papers are published are listed. Which objective or research question does this information answer/pertain to?

Response: This listing of journals is typical of many of the literature reviews in the geography, social sciences, environmental and ecology literature. But we agree that this information is not central to the study aims. We have therefore removed the table (Table 2) from the manuscript and have placed it as Supporting Information Table 1. We have still left the text in the manuscript as it is relevant under the Performance analysis and sample literature characteristics section, to help characterize the literature.

Table 2. I am very surprised that journals such as Environmental Health Perspectives, Environmental Health, Environmental Research are not included in this list. Each journal contains far more than 6 articles on health-related impacts of wildfire.

Response: As explained above, we did not include the health aspects of wildfire smoke and/or exposure. The top four journals for wildfire smoke research appear to be: Environmental Health, Environmental Health Perspectives, Environmental Toxicology and Pharmacology, and International Journal of Environmental Research and Public Health. These journals are all in the public, environmental & occupational health field, which is a separate and very different audience that we are aiming for with this manuscript as outlined above. Thus, those journals did not show up in Table 2.

It is difficult to understand the need, and use for the environmental justice question and results. 

Response: This has been addressed in our response to the Editor and Reviewer 1’s, 6-7th comments.

The authors have described the aspects of environmental justice that are addressed in human impacts of wildfire literature, but give no information about content, which would be more useful. A more useful question might be: what aspects of human impacts of wildfire does EJ literature highlight? Or perhaps what distinguishes EJ literature on human impacts of wildfire from the other non-EJ literature? The authors could give some categories and some examples.

Response: Thank you for your comments. Regarding the first question, in the results section we identify the most common variables associated with papers designated as EJ, “The WUI and rural - WUI designated community types decreased the likelihood that a study was EJ relevant. Although the type of hazard studied was not significant, papers examining the post-wildfire period were more likely to be EJ than those considering another time period. In terms of sociodemographic characteristics, only race/ethnicity and poverty were significant, both of which increased the likelihood that a study is EJ (Table 6). Chi-square tests of the relationship between EJ and sociodemographic variables also found that both race/ethnicity and poverty were highly significant (p < 0.01). Additionally, employment and gender (both p < 0.50), and language (p < 0.10) were also statistically significant.” (lines 470-479).

Line 430: “Our findings identified disparities in the foci of publications in terms of effects, community types, and time periods,” I did not see any analyses or reporting of “effects” found in the selected studies.

Response: Apologies. Please see SI Table 4 and Section 2.2 where we interchangeably use “hazards” and effects”. We have now revised this; and we specifically refer to our effects variables see Lines 328-330, “H_i is a set of variables that contain the kinds of effects the paper considered such as: wildfire, smoke, fire effects, etc.”

Discussion

“Conversely, human dimensions literature focused on events occurring during (e.g., evacuation) and after wildfires (e.g., recovery and adaptation) were comparatively less studied.” It is difficult for me to believe this. There may be hundreds if not thousands of studies on the social, economic and health impacts of wildfire. I cannot tell why the authors came to this conclusion because the selected literature are not listed anywhere in the paper, and I do not know which databases the authors searched.

Response: We disagree. Please see our above responses to you regarding our aim and objectives, methods, and databases used. Furthermore, our research objective-driven study is not, and should not be, a result of “believing this”, or speculation, but rather findings based on our scientific methods. Accordingly, we believe we have done this, addressed your comments, justified that our methods are transparent and repeatable and that our findings, before and after revision requests by the other 3 reviewers, are indeed valid. 

However, we do note that in 2022 the United Nations Environment Program found that, “more than half the expenditures related to wildfires are for response, while planning typically receives just 0.2 per cent of the total budget for wildfires….. to reduce the outsized costs from damage and loss – which greatly exceeds all spending on wildfire management – we need to rebalance our efforts. …as a starting point, countries may consider rebalancing investments by up to 1 per cent for planning, 32 per cent for prevention, 13 per cent for preparedness, 34 per cent for response, and up to 20 percent for recovery.” (Page 6: United Nations Environment Programme (2022). Spreading like Wildfire – The Rising Threat of Extraordinary Landscape Fires. A UNEP Rapid Response Assessment. Nairobi.

For this reason, further statements or conclusions about “amount of research” on various topics, community types, and EJ-focused studies, are suspect.

Response: We disagree, and it is unfortunate you reached this conclusion. But please see all the previous responses to your comments. We note that the 3 other reviewers’ comments, regarding this issue, were very different from yours.

If the authors included only English-language publications, then it is not worth assuming that there has been no research on social dimensions of wildfires outside of English-speaking countries.

Response: To clarify, we do not assume that there has been no research on this topic outside English-speaking countries. In the original manuscript, we clearly state that this is a limitation of this paper, “First, we note that our review methods omitted some relevant global literature since it was limited to English language publications indexed in available databases. This means that that many relevant publications from frequently wildfire affected Mediterranean and Latin American regions were likely excluded.” (lines 632-635).

It is difficult to tell what the purpose of the EJ discussion is. For example: “Also, overall EJ relevant studies significantly focused on race/ethnicity and poverty aspects and to a lesser degree on employment, gender, and language (Supporting Information 4).”Race/ethnicity and poverty aspects of what? Disaster planning? Social and economic impacts of wildfire?

Response: Please see all our previous responses and revisions (including new paragraph of text) we have made to incorporate your and the other 3 reviewer’s comments. We believe we have elucidated what the purpose of the EJ discussion is.

---

## [Decision Letter · Decision Letter 1]

19 May 2022

PONE-D-22-00418R1A burning issue: Reviewing the socio-demographic and environmental justice aspects of the wildfire literaturePLOS ONE

Dear Dr. Escobedo,

Thank you for submitting your manuscript to PLOS ONE. After careful consideration, we feel that it has merit but does not fully meet PLOS ONE’s publication criteria as it currently stands. Therefore, we invite you to submit a revised version of the manuscript that addresses the points raised during the review process. Please respond to the summary suggestions as well as those of the individual reviewers.

We look forward to receiving your revised manuscript.

Kind regards,

Julia A. Jones

Academic Editor

PLOS ONE

Journal Requirements:

Additional Editor Comments:

Please make the following changes to your paper:

1) your findings, and the reviewer comments, seem to point to several important findings that could be more clearly communicated in the abstract, introduction, and discussion. These relate to the four connected themes you addressed: wildfires, the wildland-urban interface, environmental justice, and health. (1) Over the past few decades, wildfire behavior has changed: formerly wildfires tended to be confined to natural vegetation (forests, scrublands, grasslands), but increasingly they are impacting residential rural, peri-urban, and urban areas. (2) This means that wildfire impacts on people are no longer confined to the wildland-urban interface, defined as a zone far removed from urban areas, and instead wildfires are affecting rural, suburban, and urban areas. (3) In turn, this means that multiple different groups of vulnerable people are being affected by wildfire, including (a) poor, white, underinsured people living in depressed rural towns such as former mining, forestry, or other areas, (b) marginalized groups living in peri-urban or urban settings where dwellings are vulnerable, such as migrant worker trailer parks, and (c) marginalized groups living in urban areas who may be disproportionately affected by smoke from wildfires because of a lack of capacity to adapt to these conditions. (4) Thus, wildfire now poses a wide range of threats to livelihood, life, and health across all rural, suburban, and urban areas. The literature that you review does not appear to have yet adapted to this new reality, and it would strengthen your paper if you could emphasize these gaps and the need for more work in these topics.

2) your manuscript is currently almost 10,500 words long. To increase the readability and impact of your paper, please shorten it to 6000-7000 words. Some portions of text could be moved into a supplement.

3) I note that the text contains a Table 1, Table 2, and Table 6. Please adjust/correct, and check the figure numbers also.

If you are able to make these changes, it may not be necessary to send your paper out for a third round of reviews.

Reviewers' comments:

Reviewer's Responses to Questions

**Comments to the Author**

1. If the authors have adequately addressed your comments raised in a previous round of review and you feel that this manuscript is now acceptable for publication, you may indicate that here to bypass the “Comments to the Author” section, enter your conflict of interest statement in the “Confidential to Editor” section, and submit your "Accept" recommendation.

Reviewer #2: All comments have been addressed

Reviewer #3: (No Response)

Reviewer #4: (No Response)

2. Is the manuscript technically sound, and do the data support the conclusions?

Reviewer #2: Yes

Reviewer #3: Yes

Reviewer #4: Yes

3. Has the statistical analysis been performed appropriately and rigorously? 

Reviewer #2: I Don't Know

Reviewer #3: I Don't Know

Reviewer #4: Yes

4. Have the authors made all data underlying the findings in their manuscript fully available?

Reviewer #2: Yes

Reviewer #3: Yes

Reviewer #4: No

5. Is the manuscript presented in an intelligible fashion and written in standard English?

Reviewer #2: Yes

Reviewer #3: Yes

Reviewer #4: Yes

6. Review Comments to the Author

Reviewer #2: I am confused as to how the dendrogram in Supplemental Figure 1 negates the need for any mention of health literature. While it clearly shows the distinction between the air pollution topics and socio-demographic topics, many of the socio-demographic categories are related to health and should be treated as such. To define health as only having to do with the branch of air quality and wildfires is not accurate. The other reasonings presented to not include health were more compelling (saturation of the literature), however it still seems to be an oversight to omit health searches when they could be directly related to some of the other socio-demographics topics discussed. This should be mentioned in the discussion.

Reviewer #3: In my original comments, I asked for more of an explanation of how EJ factors make people more vulnerable. Based on your response, it seems the primary groups impacted are Native Americans and sometimes the elderly. Most EJ groups don't live in fire prone areas, so the application of an EJ framing seems a bit of a stretch. The authors need to state more explicitly how EJ does affect Indians and elderly people, given they are more at risk.

Which authors are referenced here: “Conversely, these authors note that lower income and minority populations have been largely reported to be concentrated in areas historically at lower risk, such as in city cores and highly urbanized suburbs.”

Amenity migration by higher income populations to peri-urban areas also puts existing residents, that are often vulnerable populations, at risk in those communities through a larger number of homes to defend in case of wildfire, since newer residents often choose not to engage in fuel reduction projects

(Collins, 2008).

“through a….” needs to be reworded. Not the correct term

However, we respectfully disagree with you regarding the use of “choices” and “positive terms”. The definition of Environmental Justice as used in this paper is a framework to understand the inherent lack of choices, and limitations, encountered by historically disadvantaged communities.

Again, this framing is too absolutist. People always have choices. You suggest that this cannot be the case ever, when in fact there is always some degree of agency. Your definition of EJ should be tempered.

Reviewer #4: The manuscript is much improved. However there are some opportunities for further improvement.

While the authors seem to have taken my questions and suggestions as personal attacks, my suggestions were aimed at improving the manuscript so that it could reach and speak to a broader audience, and thereby have an improved impact. I did not appreciate the tone of the authors responses to my comments. For example, my observation of use of forestry jargon could be interpreted to mean that the non-forestry audience reading PlosOne would not comprehend the content of this manuscript. PlosOne is not a forestry journal, in fact it is explicitly a multi-disciplinary journal, and therefore publishing in this journal is an opportunity to reach broader audiences. I suggest that the authors complete their revision/response, put it aside, and then return to it with an eye towards whether they have responded in a constructive and respectful manner. Another option is to ask an experienced colleague to review their response document prior to submitting.

I do not believe that the authors’ response to my comment about lacking definitions of terms such as WUI was adequate. The authors suggest the reader go read another paper to understand what WUI is and why it’s important. If the authors want to only speak to a forestry audience, why do they not publish this piece in a forestry journal?

Instead, the authors could add a few sentences stating what the WUI is, the population that lives there, and that they are relatively more vulnerable to impacts from natural disasters such as wildfires.

My other major concern is that the manuscript is very long, and the writing is not concise. The text could be significantly cut (perhaps by 30-40%) with attention to conciseness. I defer to the editor on this issue.

7. PLOS authors have the option to publish the peer review history of their article (what does this mean?). If published, this will include your full peer review and any attached files.

Reviewer #2: No

Reviewer #3: No

Reviewer #4: No

---

## [Author Response · Author response to Decision Letter 1]

17 Jun 2022

RESPONSE TO REVIEWERS

Additional Editor Comments:

Please make the following changes to your paper:

1) your findings, and the reviewer comments, seem to point to several important findings that could be more clearly communicated in the abstract, introduction, and discussion. These relate to the four connected themes you addressed: wildfires, the wildland-urban interface, environmental justice, and health. 

• Over the past few decades, wildfire behavior has changed: formerly wildfires tended to be confined to natural vegetation (forests, scrublands, grasslands), but increasingly they are impacting residential rural, peri-urban, and urban areas. 

• This means that wildfire impacts on people are no longer confined to the wildland-urban interface, defined as a zone far removed from urban areas, and instead wildfires are affecting rural, suburban, and urban areas. 

• In turn, this means that multiple different groups of vulnerable people are being affected by wildfire, including (a) poor, white, underinsured people living in depressed rural towns such as former mining, forestry, or other areas, (b) marginalized groups living in peri-urban or urban settings where dwellings are vulnerable, such as migrant worker trailer parks, and (c) marginalized groups living in urban areas who may be disproportionately affected by smoke from wildfires because of a lack of capacity to adapt to these conditions. 

• Thus, wildfire now poses a wide range of threats to livelihood, life, and health across all rural, suburban, and urban areas. The literature that you review does not appear to have yet adapted to this new reality, and it would strengthen your paper if you could emphasize these gaps and the need for more work in these topics.

Thank you for summarizing our study. Indeed, we have taken these 4 themes to heart and have reorganized our Discussion and much of our Conclusion into 3 sections. The first section is where we discuss our findings relative to our 3 stated objectives. This has served to focus our discussion and to not tangentially delve into disparate topics. We then have a distinct limitations paragraph. And we then have reorganized and presented the end of our discussion text around a distinct Future Research section which is organized according to the 4 themes your highlighted. Finally, we close with a short succinct 2 paragraph Conclusion. As you will see we have also revised our Introduction following your indications and that of other reviews. We have also emphasized the use of our proposed wildfire relevant EJ definition in formulating environmental policies. 

2) your manuscript is currently almost 10,500 words long. To increase the readability and impact of your paper, please shorten it to 6000-7000 words. Some portions of text could be moved into a supplement.

We have eliminated repetitive text, nonrelevant information and moved entire sections to Supporting Information. This has reduced the word count in the main text (i.e., Introduction to Conclusion) to 7, 338 words. We have also eliminated 12 references that were comprised of approximately 340 words (these references were unnecessary or repeating other references). As requested, we have also moved Background on US policy on Environmental justice, literature review categories, and bibliometric and statistical analyses methods; plus journal and geographical scope, contexts, and collaborations results to the Supporting Information sections. We have also consolidated and reduced the number of supporting information files. 

3) I note that the text contains a Table 1, Table 2, and Table 6. Please adjust/correct, and check the figure numbers also.

We apologize for this mistake. This is in large part due to our use of Supporting Information materials. However, we have changed Table 6 to Table 2:

“both of which increased the likelihood that a study was EJ (Table 2)” lines 368-369

“Table 2. Results of the probit model” line 375

“have a lower likelihood of being labeled as EJ (Table 2).” Line 487

We have also double checked the other figure, table, and text numbers in the text and Supporting Information section for consistent and accurate numbering.

If you are able to make these changes, it may not be necessary to send your paper out for a third round of reviews.

Comments to the Author

6. Review Comments to the Author

Reviewer #2: I am confused as to how the dendrogram in Supplemental Figure 1 negates the need for any mention of health literature. While it clearly shows the distinction between the air pollution topics and socio-demographic topics, many of the socio-demographic categories are related to health and should be treated as such. To define health as only having to do with the branch of air quality and wildfires is not accurate. The other reasonings presented to not include health were more compelling (saturation of the literature), however it still seems to be an oversight to omit health searches when they could be directly related to some of the other socio-demographics topics discussed. This should be mentioned in the discussion.

You are correct in that health issues do influence social vulnerability. There are two subsets of the wildfire smoke literature. The first relates to exposure to the smoke, either wildfire or prescribed fire. The second would be the health effects of the smoke which is a different field targeted towards a different audience and published in different journals (as we previously mentioned). However, we agree with your suggestion to clearly link health and other socio-demographic topics. We have added the following text, “Similarly, certain socio-demographic groups are more vulnerable to wildfire smoke effects due to factors such as pre-existing health conditions, and a lack of insurance/medical care. In Australia, effects of wildfire smoke on health tended to be stronger for Aboriginal populations (Hanigan et al., 2008).” lines 409-412. We have added a new reference accordingly (D’Evelyn et al., 2022) to further substantiate the saturation of the literature issue you raise. 

 Reviewer #3: In my original comments, I asked for more of an explanation of how EJ factors make people more vulnerable. Based on your response, it seems the primary groups impacted are Native Americans and sometimes the elderly. Most EJ groups don't live in fire prone areas, so the application of an EJ framing seems a bit of a stretch. The authors need to state more explicitly how EJ does affect Indians and elderly people, given they are more at risk.

Thank you for providing the opportunity to address this point. It is true that two US communities that are at risk of wildfire related effects are the elderly and Native American groups; however, we argue that there are other socio-demographic groups across the globe that are affected by wildfire than those two. And this will depend on geographies, context, countries, and regions within those countries. We contend that EJ communities are present in fire prone areas –not only the US but across the globe- and that the number of EJ communities at risk continue to grow. In order to clarify all of these points and to address this comment, we added and edited language on Page 23, which now reads:

“Overall, studies from our review that focused on EJ described different impacts to, and the social vulnerability of different communities. For example, Masri et al. (2021) found that except for Native American communities, wildfires impact census tracts with lower populations of minorities and higher populations of elderly. Elderly populations are often dependent on external assistance for evacuation (Masri et al., 2021) and are also highly susceptible to smoke-related health conditions (Liu et al., 2015). Furthermore, the elderly population is more likely to have fixed incomes, making them more likely to relocate to peri-urban zones due to housing price increases (Murphy and Allard, 2015), and have limited capacity to address the financial hardships of a wildfire. In addition to being disproportionately located in areas of high fire risk, Native American communities also have high social vulnerability and low adaptive capacity (Davies et al., 2018); including high levels of poverty, low levels of education, and commonly experience household crowding.” lines 488-499

Which authors are referenced here: “Conversely, these authors note that lower income and minority populations have been largely reported to be concentrated in areas historically at lower risk, such as in city cores and highly urbanized suburbs.”

Wigtil et al. (2016) and Davies et al. (2018) as stated in the preceding sentence; hance “..these authors..”. 

……Amenity migration by higher income populations to peri-urban areas also puts existing residents, that are often vulnerable populations, at risk in those communities through a larger number of homes to defend in case of wildfire, since newer residents often choose not to engage in fuel reduction projects

(Collins, 2008).

Collins, 2008 is referred to here as cited at the end of the text you mention. 

“through a….” needs to be reworded. Not the correct term

We have rephrased this and it now reads, “…rather than strictly through a land, resource, or fire management lens..”

However, we respectfully disagree with you regarding the use of “choices” and “positive terms”. The definition of Environmental Justice as used in this paper is a framework to understand the inherent lack of choices, and limitations, encountered by historically disadvantaged communities. 

Again, this framing is too absolutist. People always have choices. You suggest that this cannot be the case ever, when in fact there is always some degree of agency. Your definition of EJ should be tempered.

Thank you for your comments. We understand that there is always some degree of agency when studying human populations. But is “choice” really viable when considering injustices and people with no power or influence, and historical legacies of disenfranchisement? One could argue that people always can move to another home or area, but the reality is that many people cannot. Perhaps they cannot afford to move because all other areas near their employment are too expensive. Perhaps they depend on nearby family who live next door for childcare. Native Americans did not choose to live on reservations and given their high poverty and strong place attachment, where would they move to? There are many reasons why one simply cannot choose to move to a place at less risk. The very people who have more choices are usually the ones that are not highly vulnerable. We also note that nowhere in our definition does it say that these discrepancies are due to a lack of choices. “Inequitably” references to the outcome, not the process. 

However, to address your point, we conclude our study, by stating, “In conclusion, it is our hope that our proposed definition of EJ can be used by both governmental and non-governmental organizations to guide the inclusion of EJ in their fire, land, and environmental policies and management efforts. However, care is warranted in that EJ should not be used as a quantitative or binary metric. Rather, EJ can be better used as a lens or framework to understand the impacts of emerging environmental problems affecting diverse and often disadvantaged communities.” Lines 631-636

Reviewer #4: The manuscript is much improved. However there are some opportunities for further improvement.

While the authors seem to have taken my questions and suggestions as personal attacks, my suggestions were aimed at improving the manuscript so that it could reach and speak to a broader audience, and thereby have an improved impact. I did not appreciate the tone of the authors responses to my comments. For example, my observation of use of forestry jargon could be interpreted to mean that the non-forestry audience reading PlosOne would not comprehend the content of this manuscript. PlosOne is not a forestry journal, in fact it is explicitly a multi-disciplinary journal, and therefore publishing in this journal is an opportunity to reach broader audiences. I suggest that the authors complete their revision/response, put it aside, and then return to it with an eye towards whether they have responded in a constructive and respectful manner. Another option is to ask an experienced colleague to review their response document prior to submitting.

We apologize for appearing to take this as a personal attack. We did not I assure you. What we were doing is simply responding in-kind to the tone you were using. For example, we know that PLOS One is not a forestry journal; in fact, most of the interdisciplinary co-authors have previously published before in this journal. And having used the term critical flaw and questioning the transparency and repeatability of our methods; also made us respond in an assertive manner since our scientific integrity was being questioned. 

That said, we apologize and assure you that in the future we will be more moderated in our response to reviewers. Alas, we do thank you and apologize for this unnecessary exchange. Your comments did improve our manuscript and we are extremely grateful. Again, apologies for any misunderstandings. 

I do not believe that the authors’ response to my comment about lacking definitions of terms such as WUI was adequate. The authors suggest the reader go read another paper to understand what WUI is and why it’s important. If the authors want to only speak to a forestry audience, why do they not publish this piece in a forestry journal?......Instead, the authors could add a few sentences stating what the WUI is, the population that lives there, and that they are relatively more vulnerable to impacts from natural disasters such as wildfires.

Interestingly WUI is definitely “forestry jargon” which you requested we avoid. However, we have now included several sentences in the Introduction to define this area and why it is important: “The WUI is defined as ““the area where houses exist at more than 1 housing unit per 40 acres and (1) wildland vegetation covers more than 50% of the land area or (2) wildland vegetation covers less than 50% of the land area, but a large area (over 1,235 acres) covered with more than 75% wildland vegetation is within 1.5 miles” (Stewart et al., 2007, p.204). This area is important in wildfire management for two reasons: 1) due to the human population there will be more human caused ignitions and thus wildfires, and 2) wildfires occurring in this area pose a greater risk to structures and lives than one in the forest or more rural areas, are harder to fight, and cannot be left to burn (Radeloff et al. 2018).” Lines 82-89. We have added a new reference accordingly. 

My other major concern is that the manuscript is very long, and the writing is not concise. The text could be significantly cut (perhaps by 30-40%) with attention to conciseness. I defer to the editor on this issue.

Please see our responses to the editor’s requests regarding these same issues. I believe you issues have now been addressed.

---

## [Editor Report · Decision Letter 2]

22 Jun 2022

A burning issue: Reviewing the socio-demographic and environmental justice aspects of the wildfire literature

PONE-D-22-00418R2

Dear Dr. Escobedo,

We’re pleased to inform you that your manuscript has been judged scientifically suitable for publication and will be formally accepted for publication once it meets all outstanding technical requirements.

Kind regards,

Julia A. Jones

Academic Editor

PLOS ONE

Additional Editor Comments (optional):

Thank you for addressing all reviewer comments and responding to my comments and suggestions.
---

## [Editor Report · Acceptance letter]

19 Jul 2022

PONE-D-22-00418R2 

A burning issue: Reviewing the socio-demographic and environmental justice aspects of the wildfire literature 

Dear Dr. Escobedo:

I'm pleased to inform you that your manuscript has been deemed suitable for publication in PLOS ONE. Congratulations! Your manuscript is now with our production department. 

Kind regards, 

on behalf of

Dr. Julia A. Jones 

Academic Editor

PLOS ONE